# p53 pulse modulation differentially regulates target gene promoters to regulate cell fate decisions

Marie D Harton[1,†], Woo Seuk Koh[1,†], Amie D Bunker[1,†], Abhyudai Singh[2] & Eric Batchelor[1,†,*] [ID]

## Abstract

The p53 tumor suppressor regulates distinct responses to cellular stresses. Although different stresses generate different p53 dynamics, the mechanisms by which cells decode p53 dynamics to differentially regulate target genes are not well understood. Here, we determined in individual cells how canonical p53 target gene promoters vary in responsiveness to features of p53 dynamics. Employing a chemical perturbation approach, we independently modulated p53 pulse amplitude, duration, or frequency, and we then monitored p53 levels and target promoter activation in individual cells. We identified distinct signal processing features—thresholding in response to amplitude modulation, a refractory period in response to duration modulation, and dynamic filtering in response to frequency modulation. We then showed that the signal processing features not only affect p53 target promoter activation, they also affect p53 regulation and downstream cellular functions. Our study shows how different promoters can differentially decode features of p53 dynamics to generate distinct responses, providing insight into how perturbing p53 dynamics can be used to generate distinct cell fates.

**Keywords** gene regulation; microfluidics; p53; single cell; transcription factor dynamics

**Subject Categories** Chromatin, Transcription & Genomics; DNA Replication & Repair

**Mol Syst Biol. (2019) 15: e8685**

## Introduction

Pulsatile dynamics have been identified in a growing number of important cellular signal transduction pathways. In human cells, the transcription factors p53 and NF-κB, as well as the extracellular signal-regulated kinase (ERK), are recent examples of signaling molecules shown to oscillate in protein expression level or activity (Lahav *et al*, 2004; Nelson *et al*, 2004; Batchelor *et al*, 2011; Albeck *et al*, 2013; Lee *et al*, 2014; Kellogg & Tay, 2015; Ryu *et al*, 2015). In the relatively simple organism *Saccharomyces cerevisiae*, a screen

identified 10 distinct transcription factors with pulsatile dynamics (Dalal *et al*, 2014). The dynamics can encode information about the specific stimulus for signaling molecules, as was observed for p53 in which pulse amplitude and duration change in response to different stimuli (Batchelor *et al*, 2011). While the regulatory mechanisms responsible for the stimulus-specific shaping of pulsatile dynamics have been determined for many systems (Batchelor *et al*, 2011; Hao & O'Shea, 2011), it remains a challenge to identify the mechanisms by which cells decode the dynamics to generate diverse output responses. Given the importance of many of the pulsatile systems in regulating stress responses and cell fate decisions, developing methods to precisely control the dynamics of key signaling molecules may provide novel methods for pharmacological interventions.

The p53 tumor suppressor is mutated in the majority of human cancers and is thus a robust target for cancer therapeutics. p53 responds to various stress signals and subsequently regulates several distinct cell fate pathways, including cell cycle arrest, apoptosis, and senescence (Batchelor *et al*, 2009; Vousden & Prives, 2009; Purvis & Lahav, 2013). Single-cell studies have demonstrated that p53 undergoes complex, stimulus-dependent dynamics. In response to DNA double-strand breaks, p53 levels increase in a series of discrete pulses of fixed average amplitude, duration, and frequency; in response to ultraviolet radiation, p53 levels increase in a single pulse with a dose-dependent amplitude and duration (Lahav *et al*, 2004; Batchelor *et al*, 2011). Alteration to p53 expression dynamics through pharmacological inhibition of the E3 ubiquitin ligase MDM2 directly impacts target gene expression patterns (Porter *et al*, 2016) and p53-mediated cell fate decisions (Purvis *et al*, 2012).

While altering p53 dynamics can change cell fate (Purvis *et al*, 2012), different cells in a clonal population can exhibit distinct responses to the same stress stimulus. Recent cell population-level analysis of p53 has demonstrated that, on average, cells leverage differences in mRNA half-lives relative to p53 pulse frequency to induce diverse target gene responses (Porter *et al*, 2016; Hafner *et al*, 2017); however, it is unclear how p53 pulses induce differences in target gene expression to produce variance in cell fate outcomes in individual cells within a population. Previous studies of pulsatile transcription factors from various signal transduction pathways have elucidated downstream molecular mechanisms

1 Laboratory of Cell Biology, Center for Cancer Research, National Cancer Institute, National Institutes of Health, Bethesda, MD, USA
2 Department of Electrical and Computer Engineering, Department of Biomedical Engineering, Department of Mathematical Sciences, and Center for Bioinformatics and Computational Biology, University of Delaware, Newark, DE, USA
*Corresponding author. Tel: +1-301-451-7156; E-mail: batchelore@mail.nih.gov
†This article has been contributed to by US Government employees and their work is in the public domain in the USA.

 

important for decoding temporal dynamics into diverse target gene expression patterns, including promoter activation (Hansen & O'Shea, 2013; Hao *et al*, 2013) and regulation of target gene mRNA stability (Hao & Baltimore, 2009; Porter *et al*, 2016; Zambrano *et al*, 2016; Hafner *et al*, 2017). For the yeast transcription factor Msn2, distinct modes of target promoter activation in terms of both target gene expression variability and activation threshold are encoded by the protein's pulsatile dynamics in individual cells (Hansen & O'Shea, 2013). In mammalian cells, variation in the activation of individual targets by key cell fate regulators, such as p53, is likely to be a mechanism to provide increased variability in stress responses.

In this study, we quantified changes in the activation of two canonically regulated p53 target promoters in response to independent manipulation of p53 pulse amplitude, duration, and frequency in single living cells (Fig 1A). We found that distinct p53 target promoters produce diverse promoter activation patterns even when they have comparable mRNA half-lives and are driven from the same p53 dynamical input, displaying a range of sensitivities toward amplitude or temporal modulation of p53 pulses. We identified specific signal processing characteristics that distinguish the p53 target promoters, including amplitude thresholding and dynamic frequency filtering, and we showed p53 pulse duration sets a refractory period for re-initiation of a p53 response. We also determined how the signal processing of each target promoter affected its gene products and downstream functions of cell cycle arrest or p53 regulation. Our study demonstrates that p53 target promoters respond to specific features of transcription factor dynamics differently in individual cells, suggesting that cells may leverage target promoter activation to provide an additional level of p53 decoding beyond mRNA instability to produce distinct target gene expression patterns that ultimately impact cell fate decisions.

## Results

### A strategy to modulate p53 expression dynamics and simultaneously track target promoter activation

To determine the specific impact of p53 pulse characteristics on target promoter activation, independent of parallel DNA damage response pathways, we developed a method to control p53 dynamics in the absence of extrinsic DNA damage. We employed a chemical perturbation strategy based on Nutlin-3, a small molecule that binds to and inhibits the E3 ubiquitin ligase MDM2 leading to the stabilization of p53 protein levels (Fig 1B). As p53 transcriptionally upregulates MDM2, our strategy disrupts a negative feedback loop to enable direct and precise control of p53 expression to interrogate the effects of p53 dynamics on target promoter activation. Previous studies have demonstrated that Nutlin-3 can rapidly enter cells and be removed from cells through the addition of growth medium with or without Nutlin-3, respectively, ensuring a wide range of control in manipulating p53 dynamics (Purvis *et al*, 2012; Porter *et al*, 2016).

To simultaneously alter p53 expression dynamics and monitor target gene promoter activation, we engineered clonal cell lines with fluorescence-based reporters. To monitor p53 levels, we used a well-characterized clonal MCF7 breast carcinoma cell line that expresses a p53-Venus yellow fluorescent protein fusion in addition to, but at a lower concentration than, endogenously expressed wild-

type p53 (Loewer *et al*, 2010). To simultaneously quantify p53 target gene promoter activation in response to p53 expression dynamics, we engineered the MCF7 p53-Venus cell line to include a transcriptional reporter construct with a reduced p53 target promoter controlling expression of an mCherry red fluorescent protein reporter tagged with a nuclear localization sequence and PEST degradation sequence (Fig 1C). We focused our study on the promoters of two canonically regulated p53 target genes, *MDM2* and *CDKN1A,* that could be upregulated by Nutlin-3 alone in the absence of extrinsic DNA damage (Fig EV1A). These canonical target gene promoters have well-characterized p53 response elements, and their products MDM2 and p21 are associated with distinct downstream pathways, p53 regulation and cell cycle arrest, respectively. The transcripts have comparable half-lives of 2.66 and 2.79 h for *MDM2* and *CDKN1A*, respectively (Fig 1A; Porter *et al*, 2016), providing an opportunity to identify promoter-specific differences in target gene regulation independent of previously identified effects due to transcript decay rates (Porter *et al*, 2016, Hafner *et al*, 2017). Single copies of the transcriptional reporters were stably integrated in the genome of the parental cell line, as verified by qPCR, and clonal cell lines were established for each reporter.

To precisely control p53 dynamics in live cells, we used a microfluidic device to rapidly exchange growth media with and without Nutlin-3. We first produced p53 oscillations that were comparable in amplitude, duration, and frequency to the natural p53 response to DNA double-strand breaks. We exposed cells to medium containing 10 μM Nutlin-3 for 3 h followed by media without Nutlin-3 for 2.5 h over a 24-h time frame, which generated an induction of p53 at the population level comparable to that for treatment with the DNA double-strand break-inducing agent neocarzinostatin (Fig EV1B; Purvis *et al*, 2012). Using time-lapse fluorescence microscopy, we observed in each of the clonal reporter cell lines p53-Venus levels oscillating with a 5.5-h period and a maximum amplitude comparable to that achieved in response to DNA double-strand breaks (Fig 1B and D–F). To modulate p53 pulse amplitude, we exposed cells to different concentrations of Nutlin-3 (Fig 1B and Fig EV1C–F). To modulate p53 pulse frequency or duration, we modified the timing of Nutlin-3 delivery and washout (Fig 1B and Fig EV1G–L). Characterization of the clonal cell lines indicated that mCherry expression was induced from each of the two reduced promoters only upon Nutlin-3 treatment (Movies EV1–EV6). p53-Venus expression dynamics in response to Nutlin-3 treatment were comparable between the parental cell line and the *MDM2* promoter reporter cell line (Fig EV1M), indicating no discernible alteration of the functioning of the p53-MDM2 negative feedback loop with the addition of the *MDM2* promoter reporter. These results indicated that the fluorescence-based reporter cell lines were suitable for altering p53 pulse features and tracking both p53 expression dynamics and target promoter activation simultaneously in individual cells.

We systematically determined how p53 pulse modulation alters the activation of the *MDM2* and *CDKN1A* promoters. The two reporter cell lines were exposed to the six p53 pulse regimens that modulated either p53 pulse amplitude, duration, or frequency, and p53-Venus and mCherry expression were quantified by fluorescence microscopy (Fig 2A–G). We first determined the percentage of cells in the population that showed target promoter activation of at least twofold induction over basal expression ("responding cells") for each p53 pulse regimen (Fig EV2A–C). Traces of promoter activation

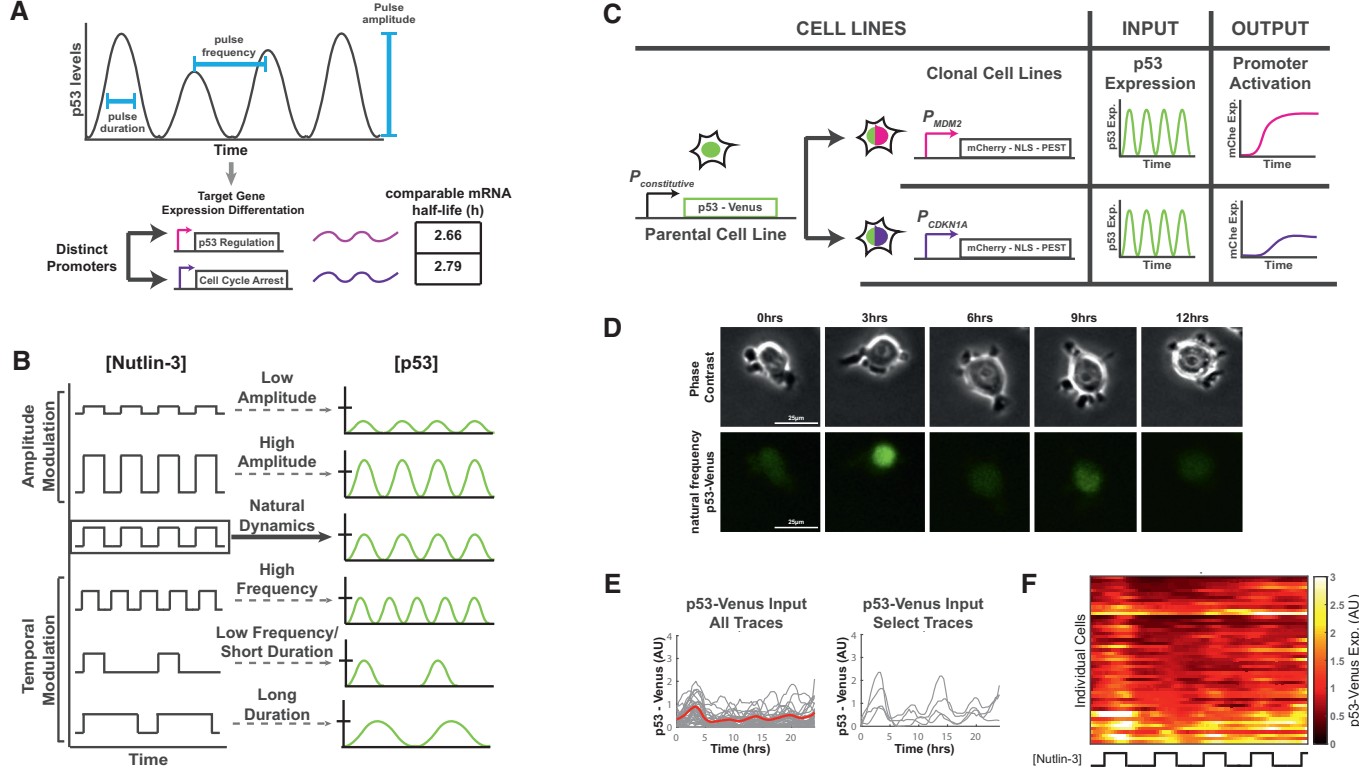

**Figure 1. A system to modulate p53 expression dynamics and monitor target promoter activation in single cells.**

A   Schematic of pulses of p53 expression in response to DNA double-strand breaks. p53 dynamics are differentially decoded to induce target genes with diverse downstream functions, including mediating cell cycle arrest and p53 regulation.

B   Schematic of modulation of p53 expression by Nutlin-3, a small molecule inhibitor that prevents p53 degradation by MDM2. Cellular exposure to media with and without Nutlin-3 potentially generates six drug dosing regimens that modulate p53 pulse amplitude, frequency, and duration.

C   Clonal cell lines expressing p53-Venus were engineered to monitor p53 expression dynamics ("input") and two canonically regulated p53 target promoters expressing mCherry ("output") in response to Nutlin-3 treatment.

D   Representative phase contrast and yellow fluorescent (indicating p53-Venus levels) images at the indicated time points for a cell exposed to the "natural dynamics" Nutlin-3 dosing regimen.

E, F   p53-Venus expression in response to the "natural dynamics" Nutlin-3 dosing regimen. Single-cell traces (gray; E) and the mean (red; E) are shown for p53-Venus expression in response to the "natural dynamics" Nutlin-3 regimen. Heat map (F) shows an alternative representation of all traces as shown in (E). N = 51 cells.

---

in individual cells were clustered based on *k*-means clustering analysis generating several distinct clusters of expression (Fig EV3), indicating a broad range of distinct promoter activation profiles across individual cells for each p53 perturbation. All cells, including non-responding cells, were considered for further analysis.

**p53 pulse amplitude delineates promoter activation thresholds**

Given the relatively complex changes in promoter activation observed upon p53 pulse modulation, we first sought to formalize our experimental results with a computational model to aid in identifying key biochemical parameters governing the promoter responses. We modeled target gene activation as a Hill function based on models previously shown to recapitulate features of other pulsatile transcription factors in terms of target promoter strength and timing (Hansen & O'Shea, 2013).

We used our simplified model to aid in determining how promoter activation is altered in individual cells in response to variation in p53 pulse amplitude. Pulsatile dynamics can be a mechanism

by which distinct activation thresholds are generated for different target genes regulated by the same transcription factor (Hansen & O'Shea, 2013). In response to DNA damage, p53 pulses have a relatively high variance (coefficient of variance of ~ 70%) within a cell and across a population of cells (Geva-Zatorsky *et al*, 2006; Toettcher *et al*, 2010). We hypothesized that individual cells leverage pulse amplitude variability to generate distinct target gene promoter activation profiles with different activation thresholds. To test this hypothesis and determine whether we could manipulate p53 pulse amplitude to activate downstream targets in a controlled manner, we analyzed the promoter response to variations in p53 amplitude across the full set of perturbations to p53 dynamics (Figs 1 and 2). At the level of individual cells, we found a high degree of variation in p53 pulse amplitude across all perturbations, resulting in a wide range of promoter activation thresholds and maximal activation levels that varied from cell to cell (e.g., Fig EV4).

To determine whether there is a more deterministic average promoter response to p53 accumulation, i.e., a relatively simple function defining the average response of a promoter to p53

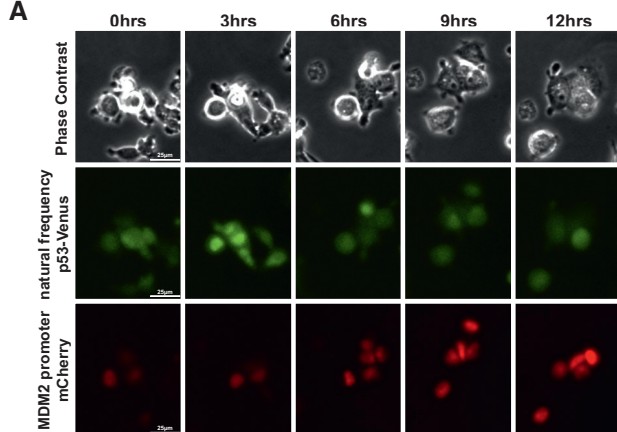

Figure 2.

**Figure 2. Characterization of target promoter activation.**

A   Representative phase contrast, yellow fluorescent (indicating p53-Venus levels), and red fluorescent (indicating reporter mCherry levels) images at the indicated time points for *MDM2* promoter reporter cells exposed to the "natural dynamics" Nutlin-3 dosing regimen.

B–G   Single-cell traces of mCherry expression for the "responding" (light gray) or "not responding" (dark gray) *MDM2* promoter or *CDKN1A* promoter cells exposed to the natural dynamics (B), low-amplitude (C), high-amplitude (D), high-frequency (E), low-frequency/short-duration (F), or long-duration (G) Nutlin-3 dosing regimens. The average trace for responding and not responding cells is shown in red and blue, respectively. Heat maps show alternative representation of all single-cell traces below each associated time course plot. *N* = at least 45 cells per condition.

concentration, we constructed the response curves for population-averaged mCherry expression from the two promoters as a function of population-averaged p53 levels. We focused on the long-duration Nutlin-3 treatment response, as it resulted in both a broad range of p53 expression levels and the highest percentage of responding cells (Fig EV2B). Fitting the fluorescence values (Figs 1 and 2) to our model of target promoter activation, we observed a monotonically increasing, nonlinear dependence between the mCherry production rate and the p53 signal (Fig 3). We found the dose–response curves were distinct for the two promoters, with the *MDM2* promoter (with fitting parameters of maximal activity $k_{max} = 135$, Hill coefficient $h = 7.5$, and threshold $K = 407$) having a lower threshold level of p53 required for activation and a higher maximal level of activation compared with the *CDKN1A* promoter (with fitting parameters of maximal activity $k_{max} = 40$, Hill coefficient $h = 7$, and threshold $K = 490$; Fig 3). These results suggest that p53 target promoters, on average, have different thresholds of promoter activation as a function of p53 pulse amplitude, potentially contributing to the diversification of target gene activation in single cells resulting from variable amplitude p53 pulses in response to DNA damage.

### p53 pulse duration establishes a refractory period for p53 pulse re-initiation

In the response to DNA double-strand breaks, there is much less variation in p53 pulse temporal characteristics—duration and

frequency—compared with the amplitude of p53 pulses (Geva-Zatorsky *et al*, 2006). This observation suggests that there may be greater selective pressure to maintain the temporal properties of p53 expression. We therefore sought to determine whether modulation of p53 pulse duration might significantly alter canonical p53 target promoter activation. We compared the response of the *MDM2* and *CDKN1A* promoters to short-duration (3 h) or long-duration (8 h) p53 pulses (Fig EV1I–L, and Fig 2F and G), characterizing promoter activation based on the time to half-maximal activation (timing), the magnitude at half-maximal activation (magnitude), and the initial rate of accumulation (rate) (Fig 4A). We found that *MDM2* promoter activation in particular was affected by an increase in p53 pulse duration, significantly increasing the magnitude (2.6-fold) and rate (1.6-fold) of promoter activation and significantly decreasing the timing (1.4-fold) (Fig 4B–G). To further test the duration sensitivity of the target promoters, we compared activation rates between the long-duration and pulsatile p53 regimens (i.e., low, natural, and high frequencies) at times corresponding to equivalent cumulative p53 levels (Fig EV5A). The *MDM2* promoter had an elevated response to the long-duration p53 input compared with all pulsatile inputs, with a greater than threefold increase in its activation rate (Fig EV5B). In contrast, the *CDKN1A* promoter had a comparable activation rate for both the long-duration and pulsatile p53 regimens (Fig EV5C). These results suggested that the *MDM2* promoter is particularly sensitive to p53 pulse duration modulation.

We next sought to identify whether the p53 pulse duration sensitivity of the *MDM2* promoter affected MDM2 function, i.e., the regulation of p53 levels. For the Nutlin-3 regimen designed to generate two long-duration p53 pulses, most cells generated either a greatly reduced p53 amplitude in the second pulse or failed to initiate a second p53 pulse altogether (Fig EV1K and L). This result suggested that a prolonged duration of p53 expression generated a refractory period in which p53 expression failed to be re-initiated. Given the increase in *MDM2* promoter activation in response to a long p53 pulse duration, we hypothesized that the refractory period was caused by excess MDM2 levels destabilizing newly synthesized p53 to the extent that it prevented a second p53 pulse from accumulating. To test this hypothesis, we monitored p53 and MDM2 protein levels in single cells using a previously characterized MCF7 clonal cell line expressing p53-CFP and MDM2-YFP fusions (Lahav *et al*, 2004) in response to a natural- or long-duration p53 pulse treatment. In cells with a long p53 pulse duration, there were significantly higher levels of MDM2-YFP at 5.5 and 11 h, corresponding to the timing of the beginning of the second and third p53 pulses, respectively, in the natural p53 response (Fig 4H). Taken together, these results suggest that repeated pulses of p53, such as those observed in response to DNA double-strand breaks, depend on pulse duration. If the duration of p53 expression exceeds the natural pulse duration,

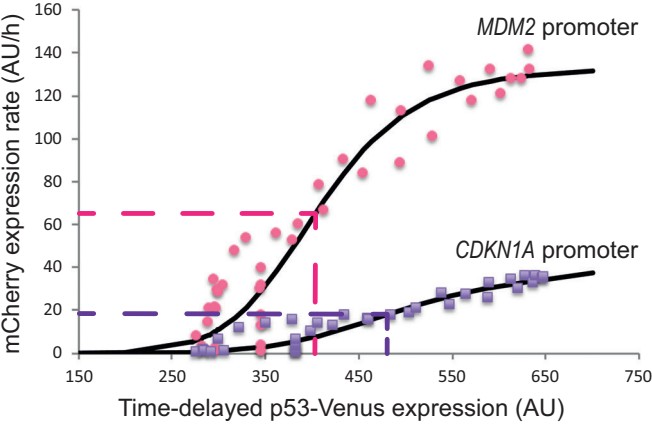

**Figure 3. p53 pulse amplitude generates promoter activation thresholds.**
Dose–response curves representing the rate of mCherry expression from the *MDM2* (pink dots) and *CDKN1A* (purple dots) promoters as a function of total p53 levels. Values of expression levels were averaged across all cells in the first 15-h response to the long-duration Nutlin-3 dosing regimen. Solid black lines represent the best fit to a Hill function model of promoter activation. Dashed lines indicate the half-maximal threshold of p53 levels for promoter activation and the half-maximal promoter activity for each promoter.

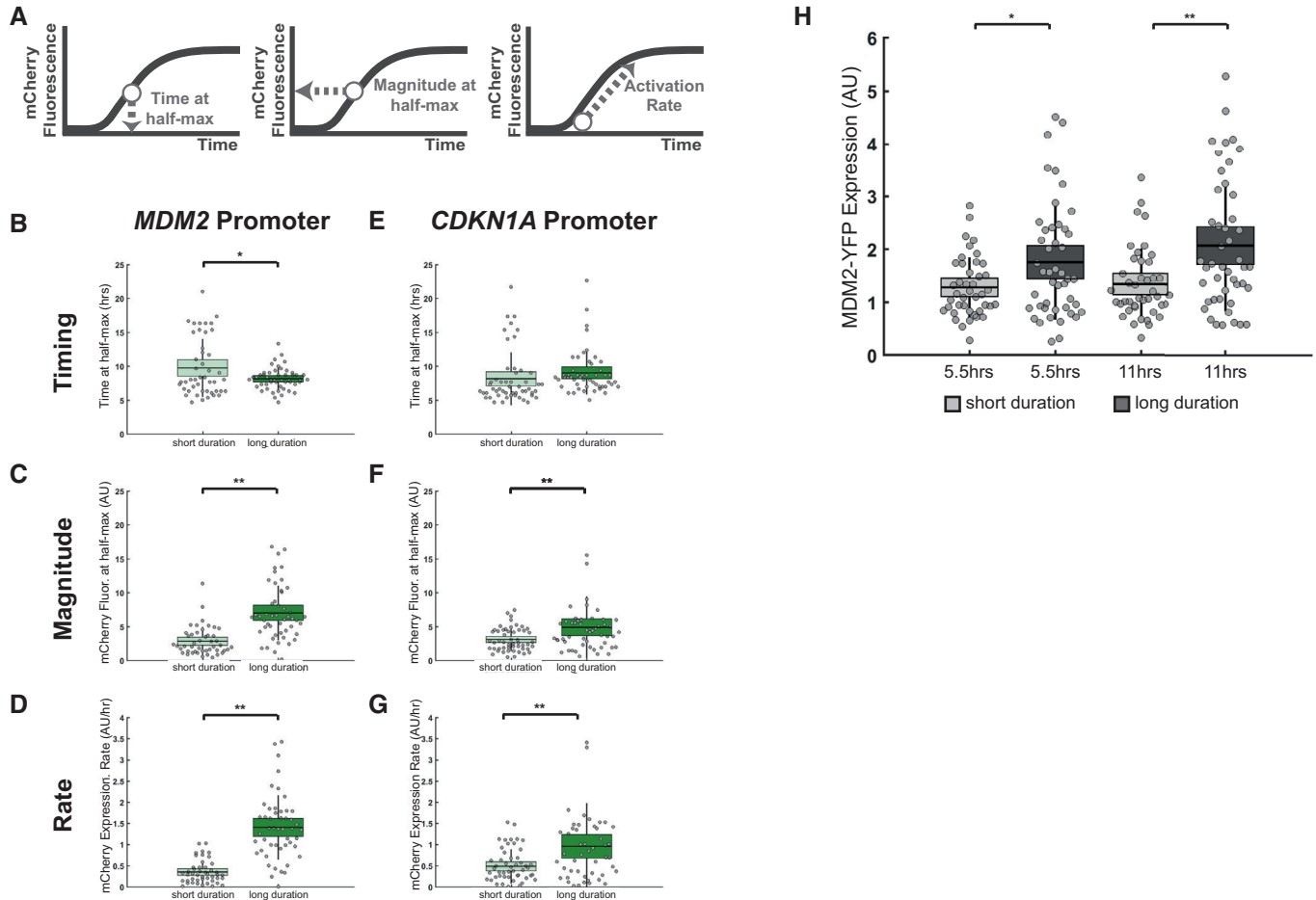

**Figure 4. p53 pulse duration establishes a refractory period.**

A    Responding cells were characterized based on the indicated metrics of target promoter activation (timing, magnitude, and rate).

B–G  Effects of p53 duration modulation on target promoter activation in terms of the mean timing (B, E), magnitude (C, F), and rate (D, G) for the *MDM2* (B–D) and *CDKN1A* (E–G) promoters in response to short (light green) and long (dark green) p53 durations. N = at least 45 cells per condition, line = mean, box = SD, bar = 95% confidence interval.

H    Mean nuclear MDM2-YFP levels in single cells at times corresponding to the first pulse (5.5 h) or second pulse (11 h) of p53 expression in response to natural- and long-duration Nutlin-3 dosing regimens. N = at least 40 cells per condition, line = mean, box = SD, bar = 95% confidence interval.

Data information: *P < 0.05 and **P < 0.01, two-sample t-test.

---

elevated MDM2 levels abrogate re-initiation of subsequent p53 pulses.

## p53 pulse frequency is differentially filtered by target promoters and controls maintenance of cell cycle arrest

The frequency of p53 pulses is relatively fixed in response to DNA double-strand breaks (Lahav *et al*, 2004; Batchelor *et al*, 2011), and it sets a timescale for defining mRNA expression dynamics as a function of transcript decay rates (Porter *et al*, 2016; Hafner *et al*, 2017). Given the sensitivity of target promoter activation to temporal modulation of p53 dynamics (Fig 4), we sought to determine whether p53 pulse frequency, similar to pulse duration, has a functional impact on target promoter activation. We first determined how modulation of the natural 5.5-h pulse frequency affected target promoter activation in terms of activation magnitude, timing, and

rate (Fig 5A–F). The magnitude and rate of *MDM2* promoter activation was highest, and the timing was fastest, at the natural p53 pulse frequency (Fig 5A–C). In contrast, the *CDKN1A* promoter was relatively unaffected by low-frequency p53 pulses compared with the natural pulse frequency; however, higher p53 pulse frequencies reduced the *CDKN1A* activation rate and magnitude and increased the timing (Fig 5D–F). These results suggest that the conserved p53 pulse frequency observed in the DNA double-strand break response is important for maintaining optimal activation of select p53 target promoters.

The fact that distinct promoters had different responsiveness to changes in p53 pulse frequency suggested that target promoters dynamically filter p53 pulses. Analogous to electrical circuits, previous studies have demonstrated that biological oscillators can differentially filter oscillating inputs (Toettcher *et al*, 2013; Mitchell *et al*, 2015). Potential classes of dynamic filters for biological

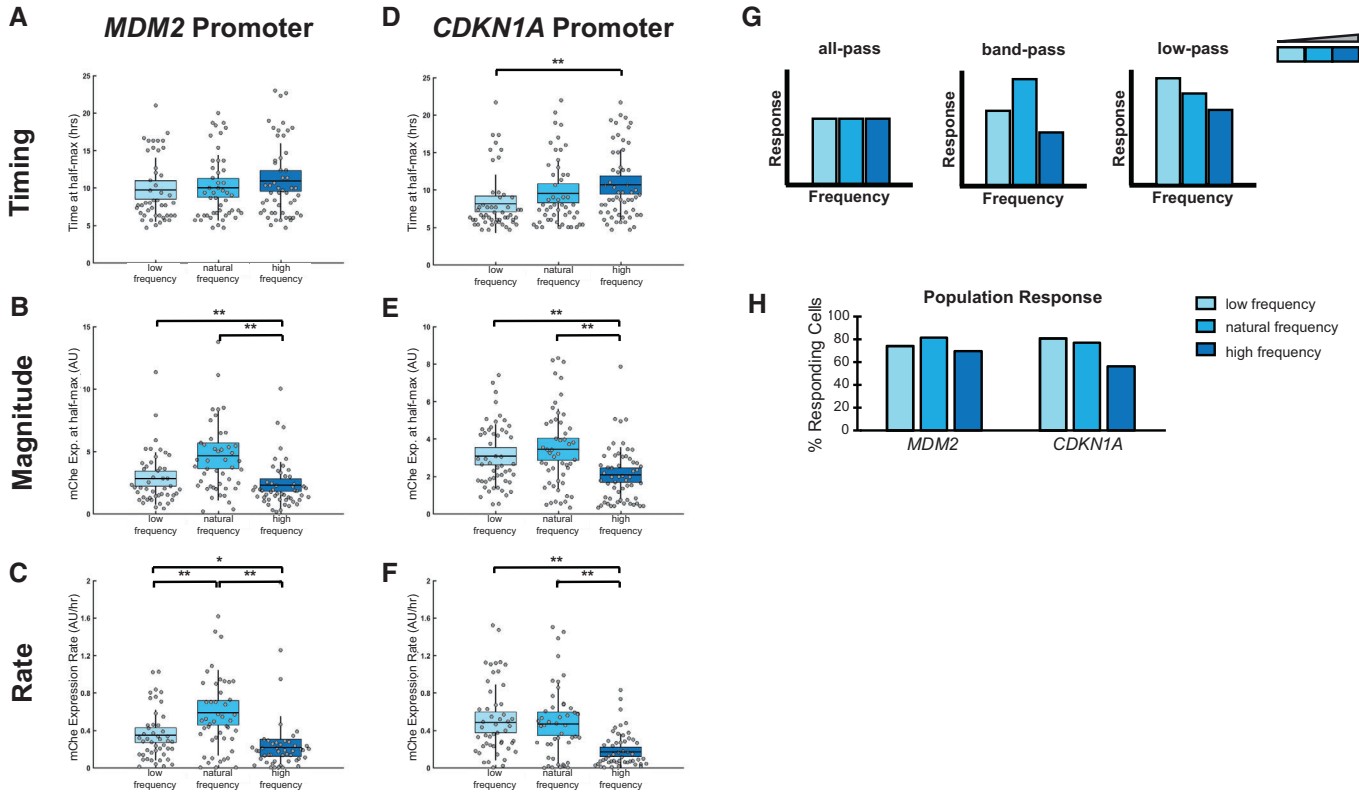

**Figure 5. p53 pulse frequency is differentially filtered by target promoters.**

A–F    Effects of p53 frequency modulation on target promoter activation in terms of the mean timing (A, D), magnitude (B, E), and rate (C, F) for the *MDM2* (A–C) and
         *CDKN1A* (D–F) promoters in response to low (light blue), natural (medium blue), and high (dark blue) p53 frequencies. *N* = at least 45 cells per condition,
         line = mean, box = SD, bar = 95% confidence interval. **P* < 0.05 and ***P* < 0.01, two-sample *t*-test.
G       Theoretical responses for all-pass, band-pass, and low-pass filters.
H       Percentage of responding cells for the *MDM2* and *CDKN1A* promoters in response to low-, natural-, and high-frequency p53 pulses. Data also shown in Fig EV2A.

systems include the low-pass filter (which is most responsive to frequencies below a specific cutoff), the band-pass filter (which is most responsive to a specific range of frequencies), and the all-pass filter (which is responsive to a wide range of frequencies; Fig 5G). Based on their responses to p53 frequency modulation, the *MDM2* and *CDKN1A* promoters showed characteristics of different filters. The *MDM2* promoter was selective for the natural p53 pulse frequency, suggesting it acted as a band-pass filter, especially in terms of the magnitude and rate of promoter activation (Fig 5B, C, and G); in contrast, the magnitude and rate of *CDKN1A* promoter activation was most responsive to low and natural frequencies, indicating low-pass filtering (Fig 5E–G). Interestingly, the filtering class of the target promoters correlated with the overall population response, i.e., the percentage of cells responding to a given p53 dynamic input. The percentage of cells showing any activation of the *MDM2* promoter was highest for the natural frequency (Fig 5H), and the percentage for the *CDKN1A* promoter was highest for the low or natural frequency (Fig 5H). These results suggest that the dynamic filtering of p53 pulses by target promoters may generate selectivity in the activation of downstream pathways for distinct modes of p53 dynamics.

Based on the low-pass filtering capability of the *CDKN1A* promoter, we hypothesized that effective cell cycle arrest

facilitated by upregulated p21 expression required p53 pulses to be below a cutoff frequency. To test this hypothesis, we monitored the number of cell divisions and the G1/S transition in live cells in response to low-, natural-, and high-frequency p53 pulses over 40 h. To detect the G1/S transition, we developed a stable MCF7 cell line simultaneously expressing p53 tagged with the Venus fluorescent protein and a previously characterized reporter in which geminin is tagged with the mCherry fluorescent protein (Fig 6A and B; Sakaue-Sawano *et al*, 2008). Geminin begins to accumulate at the end of the G1 phase of the cell cycle, continues to increase in expression throughout the S phase, and is degraded during the M phase, and therefore serves as a sensitive indicator of cell cycle progression via time-lapse fluorescence microscopy. In general, cells maintained cell cycle arrest more readily when p53 was expressed in low- or natural-frequency pulses than when expressed in high-frequency pulses (Fig 6C–E). Fewer than 30% of cells expressing low or natural p53 pulse frequencies underwent cell division within 40 h of imaging, compared with 58% of cells expressing high-frequency p53 pulses (Fig 6D). The percentage of cells showing either sustained or pulsatile expression of geminin-mCherry was 51% in response to a high p53 pulse frequency compared with only 38 and 19% in response to a low or natural p53 pulse frequency, respectively (Fig 6E), indicating

that a higher number of cells progressed through the cell cycle with a high p53 pulse frequency (Fig 6D). Although the low- and natural-frequency inputs generated similar promoter activation rates (Fig 5F), cells exposed to the natural p53 frequency showed a higher level of cell cycle arrest when considering both the percentage of cells undergoing division and expressing geminin-mCherry (Fig 6D and E).

Our previous results suggested that, in contrast to the *MDM2* promoter, the *CDKN1A* promoter is more sensitive to p53 pulse amplitude modulation rather than duration modulation (Figs 3 and 4). Analysis of cell cycle progression in response to p53 pulse amplitude or duration modulation showed that duration modulation had little impact on cell cycle arrest fidelity; in contrast, increasing p53 pulse amplitude increased the number of cells maintaining cell cycle arrest (Fig EV6), consistent with a greater

sensitivity of the *CDKN1A* promoter in response to fluctuations in p53 pulse amplitude (Fig 3) and recent work by others (Reyes *et al*, 2018). Taken together, these results suggest that p53 pulses of sufficiently high amplitude and sufficiently low frequency are required to properly activate *CDKN1A* expression to maintain cell cycle arrest.

## Discussion

The development of robust p53-based cancer therapeutics requires understanding how p53 expression dynamics are decoded by downstream cellular mechanisms to generate diverse cell fate decisions. Here, we examined how cells leverage differences in target promoter activation to produce distinct target gene

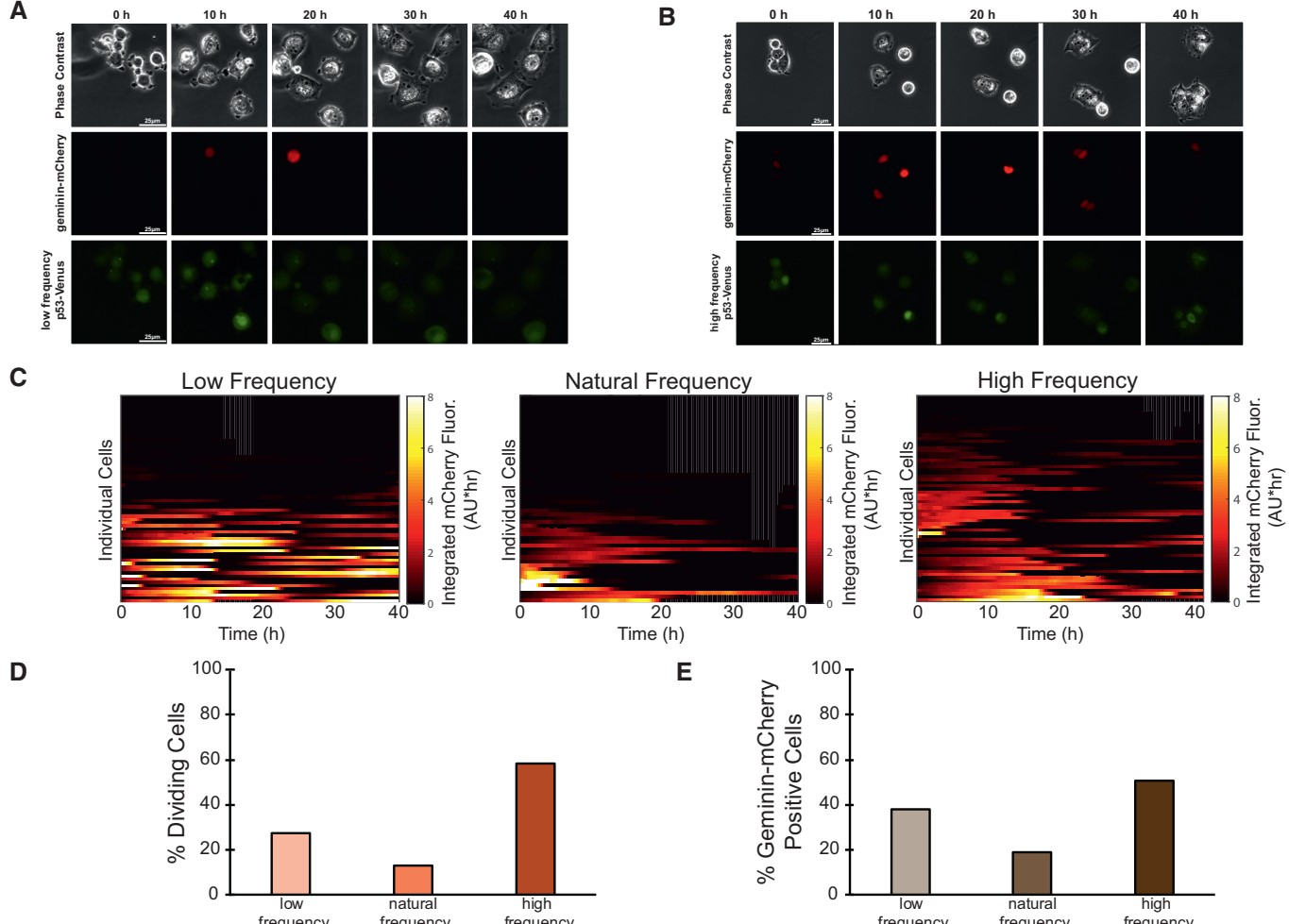

**Figure 6.  p53 frequency modulation alters cell cycle arrest.**

A, B    Representative phase contrast, red fluorescence (indicating geminin-mCherry levels), and yellow fluorescence (indicating p53-Venus levels) images of cells in response to a low-frequency (A) or high-frequency (B) Nutlin-3 dosing regimen over 40 h.

C    Geminin-mCherry traces in single cells treated with the low-, natural-, or high-frequency Nutlin-3 dosing regimens. *N* = at least 50 cells per condition.

D, E    Percentage of cells undergoing cell division (D) or positive for geminin-mCherry expression (E) within the 40 h of imaging in response to the low-, natural-, and high-frequency Nutlin-3 dosing regimens as shown in (C).

expression patterns from a single p53 pulsatile input. Employing a strategy to simultaneously modulate p53 expression dynamics and track target promoter activation in single living cells, we determined that two canonically regulated p53 target promoters undergo distinct activation modes in response to amplitude, duration, and frequency-modulated p53 dynamics. Moreover, we found that the target promoters could be differentiated based on their threshold and maximal activation in response to p53 pulse amplitude, their sensitivity to p53 pulse duration, and their filtering of p53 pulse frequency. Based on these findings, we also determined how specific p53 pulse regimens affected the primary cellular functions controlled by the *MDM2* and *CDKN1A* gene products: regulation of p53 expression and cell cycle arrest, respectively.

Previous studies of pulsatile transcription factors have identified the importance of differences in target promoter activation for generating diverse target gene expression patterns, particularly in yeast cells (Hansen & O'Shea, 2013, 2015). In these studies, four distinct promoter activation classes—characterized by a low or high amplitude threshold and fast or slow activation timescale— were computationally determined (Hansen & O'Shea, 2013, 2016). Interestingly, we found that the p53 target promoters we tested in human cells represented two of the four previously identified promoter classes based on the timescale and amplitude threshold criteria (Figs 3–5). In addition, we discovered that p53 target promoters could also be distinguished based on the strength of their temporal selection, i.e., a promoter's ability to generate elevated responses to specific temporally modulated p53 inputs as shown by promoter sensitivity to p53 pulse duration and frequency filtering (Figs 4 and 5). Although both the *MDM2* and *CDKN1A* promoters have fast activation timescales, they differ in their amplitude thresholding and strength of temporal selection. The *MDM2* promoter had a low amplitude threshold/strong temporal selection behavior while the *CDKN1A* promoter had a high amplitude threshold/weak temporal selection behavior. Thus, our focused study of p53 target promoters underscores the diversity in target promoter activation modes within the p53 signaling network, suggesting that cells leverage not only differences in mRNA half-lives (Porter *et al*, 2016), but also differences in target promoter activation to evoke distinct target gene expression patterns. Given the diversity of target promoter activation modes identified here, our study serves as a foundation for future work understanding the dynamic regulation of diverse p53-regulated genes. Promoter activation is likely to be particularly important for identifying sources of cellular heterogeneity in single-cell responses to p53-activating stimuli.

How can p53 target promoters produce different responses to the same p53 input? A recent study showed that p53 binding dynamics were similar even for target genes that clustered into different expression profiles (Hafner *et al*, 2017). Given that p53 binding dynamics are similar across target genes, our results suggest that diversity in target promoter activation may stem from differences in chromatin accessibility, sequence variability for basal transcription factor binding sites, or protein–protein interactions between the basal machinery and bound p53. Indeed, several features of target promoter sequences can greatly affect p53-mediated recruitment of basal transcriptional factors (Szak *et al*, 2001; Schlereth *et al*, 2010; Kearns *et al*, 2016;

Coleman *et al*, 2017). Future studies based on our experimental approach could employ engineered p53 target promoter sequences to systematically test how variation in either the p53 response element sequence or the spatial arrangement of core promoter features or response elements affects p53 target gene promoter activation in single cells.

The ability to filter an input signal based on its frequency is a potentially useful feature for biological signaling systems. Low-pass filtering in the Ras/Erk pathway has been shown to act as a mechanism to suppress transient fluctuations in kinase activity operating on the timescale of minutes (Toettcher *et al*, 2013). We observed low-pass filtering of p53 expression operating on the timescale of hours in the transcriptional upregulation of the *CDKN1A* promoter as well as in the fidelity of cell cycle arrest mediated by the *CDKN1A* gene product p21. These results suggest that p53 accumulation must occur for a sufficient time to effectively activate p21-mediated cell cycle arrest, potentially buffering against high-frequency transient fluctuations in p53 levels. The band-pass filtering observed for *MDM2* promoter activation suggests that both high-frequency fluctuations in p53 and persistent p53 levels are filtered to achieve maximal MDM2 expression for only a defined frequency of p53 expression. One potential mechanism by which low-frequency p53 pulsing can be filtered is through feed-forward regulation occurring for specific target promoters (Goentoro *et al*, 2009). Previous work suggested that feed-forward regulation mediated by post-translational modification of p53 in response to DNA damage may serve as a mechanism to generate a persistence detector for the activation of *CDKN1A* expression (Loewer *et al*, 2010). However, our results show that additional filtering mechanisms operate at p53 target promoters even in the absence of DNA damage. Future studies focused on perturbations of parallel p53 modification pathways or chromatin modifiers at specific target genomic loci will be required to better understand the diverse mechanisms affecting the regulation of specific p53 targets.

Our approach using chemical perturbations of p53 dynamics independent of DNA damage not only provides novel insights into the basic functioning of one of the most important stress response pathways in human cells but may also inform innovative cancer therapeutic strategies. Single-cell studies have shown that p53 pulse features can vary substantially within a cell population (Geva-Zatorsky *et al*, 2006), across cell lines (Stewart-Ornstein & Lahav, 2017), and between different species (Stewart-Ornstein *et al*, 2017). These findings increase the relevance of our chemical perturbation strategy, which enables controlled manipulation of various p53 dynamic patterns that may be relevant in different tumor environments or tissue contexts. Furthermore, our system can be combined with functional assays to determine the consequences of p53 pulse modulation on downstream cell fate effectors. Other benefits of our approach are that it uses previously developed drugs to evoke specific p53-mediated cell fate decisions and could reveal powerful co-treatment strategies with drugs that alter either p53 levels or chromatin accessibility. Strategies that repurpose currently available drugs are in high demand given the increasing difficulty to identify novel cancer targets and subsequently develop new classes of drugs (DiMasi & Grabowski, 2007; DiMasi *et al*, 2010; Hoelder *et al*, 2012; de Gramont *et al*, 2015).

# Materials and Methods

## Reagents and Tools table

| Reagent/Resource | Reference or source | Identifier or catalog number |
| --- | --- | --- |
| **Experimental models** | | |
| HEK293T (*Homo sapiens*) | ATCC | CRL-1573 |
| MCF-7 (*Homo sapiens*) | ATCC | HTB-22 |
| MCF-7 + p53-Venus | Batchelor *et al* (2008) | N/A |
| MCF-7 + p53-ECFP + *hMDM2* promoter − hMDM2 + EYFP | Lahav *et al* (2004) | N/A |
| MCF-7 + p53-Venus + *CDKN1A*p-mCherry-NLS-PEST | This study | N/A |
| MCF-7 + p53-Venus + *MDM2*p-mCherry-NLS-PEST | This study | N/A |
| MCF-7 + p53-Venus + *EF1α*p-mCherry-Geminin | This study | N/A |
| **Recombinant DNA** | | |
| pEB62 = pRRL-R4R3 + Puromycin selection marker | This study | N/A |
| pEB67 = pDONRP2RP3 + PEST-SV40 NLS | This study | N/A |
| pEB100 = pDONR221 + mCherry | This study | N/A |
| pEB78 = pDONRP4P1R + *MDM2* promoter | This study | N/A |
| pEB237 = pDONRP4P1r + *CDKN1A* promoter | This study | N/A |
| pEB330 = pEB62 + *MDM2* promoter + mCherry + PEST-NLS | This study | N/A |
| pEB331 = pEB62 + *CDKN1A* promoter + mCherry + PEST-NLS | This study | N/A |
| pLV-EF-mCherry-Geminin IRES-Puro | Cappell *et al* (2016) | N/A |
| p273 | Loewer *et al* (2010) | N/A |
| pLV-EF1a-IRES-Puro | Hayer *et al* (2016) | N/A |
| **Antibodies** | | |
| Mouse monoclonal anti-p53 DO-1 antibody | Santa Cruz Biotechnology | sc-126 |
| Rabbit monoclonal anti-β-actin 13E5 | Cell Signaling Technology | 4970 |
| Donkey anti-mouse 680RD | LI-COR Biosciences | 926-68072 |
| Donkey anti-rabbit 800CW | LI-COR Biosciences | 926-32213 |
| **Oligonucleotides** | | |
| PCR primers | This study | Table EV1 |
| **Chemicals, enzymes and other reagents** | | |
| Neocarzinostatin | Sigma-Aldrich | N9162 |
| Nutlin-3 | Sigma-Aldrich | N6287 |
| Antibiotic−antimycotic solution | Corning | 30-004-CI |
| Lipofectamine® 3000 transfection reagent | Thermo Fisher | L3000-008 |
| M-PER mammalian protein extraction reagent | Thermo Fisher | 78501 |
| SsoAdvanced™ Universal SYBR® Green Supermix | Bio-Rad | 172-5270 |
| Phusion® High-Fidelity DNA Polymerase | New England Biolabs | M0530S |
| Deoxynucleotide (dNTP) Solution Mix | New England Biolabs | N0447S |
| **Software** | | |
| MATLAB2019a | Mathworks | |
| NIS-Elements | Nikon | |
| ImageJ | Schneider *et al*, 2012; https://imagej.nih.gov/ij/ | |
| Image Studio v3.1 | LI-COR Biosciences | |
| CellASIC® ONIX2 FG | Millipore Sigma | |

**Reagents and Tools table** (continued)

| Reagent/Resource | Reference or source | Identifier or catalog number |
|---|---|---|
| CopyCaller® Software v2.0 | Applied Biosystems | |
| **Other** | | |
| TiE inverted fluorescence microscope | Nikon | |
| iXon Ultra 888 EMCCD Camera | ANDOR Technology Ltd | |
| CellASIC® ONIX2 Microfluidic System | EMD Millipore | EAR99 |
| CellASIC® ONIX Microfluidic Plates | EMD Millipore | M04S |
| MultiSite Gateway® Three-Fragment Vector Construction Kit | Thermo Fisher | 12537023 |
| Custom TaqMan™ Copy Number Assay | Applied Biosystems | 4400294 |
| TaqMan® Copy Number Reference Assay RNaseP | Applied Biosystems | 4403326 |
| Lenti-X™ Packaging Single Shots | Clontech | 631275 |
| Lenti-X GoStix™ | Clontech | 631243 |

## Methods and Protocols

### Cell line maintenance

MCF7 breast carcinoma cells expressing both the p53-Venus (Batchelor *et al*, 2008) and the promoter-mCherry fluorescent reporters were maintained at 37°C and 5% $CO_2$ in RPMI media containing 10% fetal bovine serum (FBS), 1% antibiotic–antimycotic solution (Corning 30-004-CI), 400 μg/ml neomycin (G418), and 0.5 μg/ml puromycin. HEK293T cells were maintained in DMEM containing 10% FBS and 1% antibiotic–antimycotic solution (Corning 30-004-CI).

### Microfluidic device experimental setup

The CellASIC® ONIX2 Microfluidic System (EMD Millipore, EAR99) was used to rapidly exchange media with or without Nutlin-3 to reporter cell lines. CellASIC® ONIX Microfluidic Plates (EMD Millipore, M04S) were primed with 10 μl of Transparent RPMI media containing 5% FBS and 1% antibiotic–antimycotic solution. To prepare cells for capillary-driven loading into the microfluidic plates, cells at 70–80% confluence were trypsinized and a 2 ml of aliquot was isolated and diluted into RPMI medium lacking riboflavin and phenol red (transparent RPMI) and containing 5% FBS and 1% antibiotic–antimycotic solution. The cell samples were gently vortexed, and a 10 μl aliquot was loaded into the microfluidic plates via capillary-driven loading. Standard CellASIC® ONIX2 manufacturer's protocols for capillary-driven loading were used. The cell loading process was repeated two additional times to increase the cell number within the microfluidic viewing chambers. Each well within one column of the microfluidic plate was loaded with 300 μl of transparent RPMI medium containing 5% FBS, 1% antibiotic–antimycotic solution, and 5–15 μM Nutlin-3 (N6287, Sigma-Aldrich). Each well within an adjacent column in the microfluidic plate was loaded with 300 μl of transparent RPMI medium containing 5% FBS and 1% antibiotic–antimycotic solution. The microfluidic plate was sealed to the vacuum manifold using standard CellASIC® ONIX2 manufacturer's protocols. The microfluidic plate was placed upon the plate stage and immediately imaged.

### Generation of p53-modulated pulse regimens with Nutlin-3

The CellASIC® ONIX2 FG software was used to automate the exchange of media with and without Nutlin-3. Six Nutlin-3 pulse regimens were designed: natural frequency, low frequency, high frequency, long duration, low amplitude, and high amplitude. The natural-frequency pulse regimen repeatedly exposed cells to medium with 10 μM Nutlin-3 for 3 h followed by medium without Nutlin-3 for 2.5 h. The low-frequency pulse regimen repeatedly exposed cells to medium with 10 μM Nutlin-3 for 3 h followed by medium without Nutlin-3 for 8 h. The high-frequency pulse regimen repeatedly exposed cells to medium with 10 μM Nutlin-3 for 2.5 h followed by medium without Nutlin-3 for 2 h. The long-duration pulse regimen repeatedly exposed cells to medium with 10 μM Nutlin-3 for 8 h followed by medium without Nutlin-3 for 3 h. The amplitude-modulated pulse regimens were based on the natural-frequency pulse regimen but included either 5 μM or 15 μM Nutlin-3. All programs were designed to expose activated wells within the microfluidic plate to 0.25 psi, which facilitated an average flow rate of 0.6 μl/h. Cells within the microfluidic viewing chambers were exposed to 5% $CO_2$ through the CellASIC ONIX2 system and maintained at 37°C.

### Time-lapse microscopy

Images were acquired on a Nikon TiE inverted fluorescence microscope equipped with an automatic focus correction system and the iXon Ultra 888 EMCCD Camera (ANDOR Technology Ltd). The Nikon TiE fluorescent microscope was adapted for long-term time-lapse microscopy through the addition of an environmental chamber that maintains a constant environment of 37°C, 5% $CO_2$, and 40% humidity. Images were acquired with a 20× plan apo objective (NA 0.75) every 20 min over a 24-h period. The mCherry filter set contained filters of 540–580 nm for the excitation light, 585 nm for the dichroic beam splitter, and 593–668 nm for the emission light (Chroma). The Venus filter set contained filters of 488–512 nm for the excitation light, 520 nm for the dichroic beam splitter, and 532–554 nm for the emission light (Chroma). Images were analyzed using NIS-Elements software (Nikon) and custom-written ImageJ (NIH) and MATLAB software (Mathworks), which is available upon request.

### Deterministic model

To obtain the dose responses for both target promoters, we considered the long-duration treatment that has a broad range for the p53

signal and a high percentage of cells with a promoter response (Fig EV2). We assumed that the target transcription rates monotonically increase with the p53 signal as per a Hill function, resulting in the following differential equation describing the dynamics of the mCherry protein level

$$\frac{d[\text{mCherry}]}{dt} = \frac{k_{\max} \, [\text{p53}(t - \tau)]^h}{K^h + [\text{p53}(t - \tau)]^h} - \gamma \, [\text{mCherry}]$$

where $\text{p53}(t - \tau)$ represents the delayed p53 signal, $h$ is a Hill coefficient, $k_{\max}$ is the maximum mCherry production rate, and threshold $K$ represents the p53 level at which production rate is $k_{\max}/2$. To infer the Hill function for individual cells and the average response, we used the fluorescence time courses for single-cell or the population-averaged p53-Venus and mCherry signals, respectively. Given that the promoter can become refractory to a second p53 pulse, we only considered the first 15 h of the experiment. Based on the literature, the mCherry transcription, translation, and mRNA decay rates have been measured as 2,100, 360, and 0.069 min$^{-1}$, respectively (Hinow *et al*, 2006; Milo & Phillips, 2016). The mCherry protein decay rate ($6.7 \times 10^{-4}$ min$^{-1}$) was experimentally determined in our system using a cycloheximide protein degradation assay. Based on these timescales, we ignored any appreciable decay $\gamma$ of the mCherry protein in the 15-h window of our computational analysis. The p53 signal was normalized to the total of the p53 and p53-Venus signal for each cell line as determined by Western blot analysis (Fig EV7). A value of 2 h was used for the time delay term $\tau$, which approximates the time for the transcription, translation and maturation of the mCherry reporter.

### Escherichia coli *plasmids and strains*

All molecular biology procedures in *E. coli* were performed using standard methods and/or Gateway cloning protocols (Ausubel *et al*, 1994; Katzen, 2007). *Escherichia coli* strains were grown at 37°C. Phusion® High-Fidelity DNA Polymerase and dNTPs for the polymerase chain reaction (PCR) were purchased from New England Biolabs. Oligonucleotides and gBlock® Gene Fragments were purchased from Integrated DNA Technologies. Plasmid DNA was purified using QIAprep Mini- or Midiprep Kits (Qiagen). PCR products and digested plasmids were purified by agarose gel electrophoresis and QIAquick spin columns from Gel Extraction Kits (Qiagen). Plasmid concentrations were determined by absorption at 260 nm, and all absorbance measurements were taken on a Nano-Drop™ 2000/2000c spectrophotometer. All aqueous solutions were made with distilled water purchased from LabChem Inc. For PCR, a Bio-Rad C1000™ Thermocycler was used. *Escherichia coli* transformations were performed using a Bio-Rad MicroPulser™. Restriction digests were performed per manufacture's protocols (New England Biolabs). DNA sequencing was performed by Eurofins Genomics.

### Target gene RT–qPCR expression measurements in response to Nutlin-3

Two days prior to Nutlin-3 treatment, $\sim 4 \times 10^5$ MCF-7 cells were plated on a 6-cm dish. Cells were treated with or without 10 μM Nutlin-3 for 6 h. Cells were harvested and frozen in a dry ice-ethanol bath. RNA extraction was performed using the QIAshredder

and RNAeasy kits (Qiagen). The RNA concentration for each sample was determined using a UV spectrophotometer, and 2 μg of RNA was used to prepare cDNA in a 20-μl reaction using a High Capacity cDNA Reverse Transcription Kit (Applied Biosystems, 4368814). cDNA samples were diluted by 1:500 in distilled water. Ten microliter qPCR products were prepared using 1 μl of the diluted cDNA sample and final concentrations of 0.4 μM primer mix and 1 μM SsoAdvanced™ Universal SYBR® Green Supermix (Bio-Rad, 172-5270). Target gene expression was measured in duplicate using the Bio-Rad CFX96™ Real-Time PCR Detection System. cDNA samples were subjected to the following thermocycling protocol: (i) hot start (95°C for 30 s), (ii) 40 cycles of PCR (95°C for 5 s, 55°C for 20 s), and (iii) a melt curve acquisition (55°C for 5 s with 0.5°C resolution).

### Western blotting

Two days prior to treatment, $\sim 4 \times 10^5$ MCF-7 cells were plated on a 6-cm dish. Cells were treated with either 400 ng/ml neocarzinostatin (Sigma, N9162), 5 μM Nutlin-3 (Sigma, N6287), 10 μM Nutlin-3, or 15 μM Nutlin-3 for 3 h. Cells were harvested by scraping and frozen in a dry ice-ethanol bath. Cells were lysed in M-PER Mammalian Protein Extraction Reagent (Thermo Fisher, 78501) according to manufacturer's protocol, and the concentration for each protein sample was determined by Bradford assay. Equivalent protein masses were separated by electrophoresis on 4–20% gradient gels (Bio-Rad, 456-1096) and blotted onto Immobilon-P PVDF membranes (Millipore, IPVH00010). Membranes were incubated with the following primary antibodies: mouse monoclonal anti-p53 DO-1 (Santa Cruz Biotechnology, sc-126) and rabbit monoclonal anti-β-actin 13E5 (Cell Signaling Technology, 4970). Membranes were then incubated with the following secondary antibodies: donkey anti-mouse 680RD (LI-COR, 926-68072) and donkey anti-rabbit 800CW (LI-COR, 926-32213). Membranes were imaged with a LI-COR Odyssey system (LI-COR) and quantified using LI-COR Image Studio v3.1 software. Following background subtraction, the band measurements were normalized to β-actin measurements in the same lane.

### Plasmid construction

The plasmids used in this study were constructed by Gateway® cloning protocols (Katzen, 2007). The following plasmids were constructed as entry clones via the BP reaction: pEB67, pEB100, pEB78, pEB237. The *MDM2* and *CDKN1A* target promoter sequences were PCR amplified from purified MCF-7 genomic DNA. The PEST-SV40 NLS sequence was amplified from plasmid, p273 (Loewer *et al*, 2010). The following expression plasmids were constructed via LR reactions containing the appropriate entry clones and pEB62, a destination vector modified for lentivirus production: pEB330 and pEB331. The pLV-EF-mCherry-Geminin IRES-Puro (gift from S. Cappell) was constructed using the Geminin-mCherry plasmid described in Cappell *et al* (2016), which was cloned into the pLV-EF1a-IRES-Puro backbone (Hayer *et al*, 2016).

### Cell line construction

The MCF-7 + p53-Venus + *CDKN1A*p-mCherry-NLS-PEST and MCF-7 + p53-Venus + *MDM2*p-mCherry-NLS-PEST clonal cell lines were constructed by lentiviral infection. Lentivirus production was performed according to the Lenti-X™ Packaging Single Shots

protocol. The pEB330 and pEB331 plasmids (7 μg of each) were diluted in 600 μl of distilled water and added to separate tubes of Lenti-X™ Packaging Single Shots (Clontech, 631275). The lentiviral vector DNA was transfected into HEK293T cells and incubated at 37°C and supplied with 5% $CO_2$. The lentivirus for both the pEB330 and pEB331 vectors was harvested after 48 h and filtered. Virus production was confirmed using Lenti-X GoStix™ (Clontech, 631243). The lentiviral preps were concentrated and used to infect the cEB61 cell line. Infected cells were transferred to selection media containing 400 μg/ml Neomycin and 0.5 μg/ml of Puromycin to prepare polyclonal stable cell lines. Clonal cell lines were generated from dilution of stable cells in selection media. Single clones were screened, and MCF-7 + p53-Venus+*CDKN1A*p-mCherry-NLS-PEST and MCF-7 + p53-Venus+*MDM2*p-mCherry-NLS-PEST were selected based on their high expression of the mCherry fluorescent reporter upon exposure to 10 μM Nutlin-3. The MCF-7 + p53-Venus+*EF1α*p-mCherry-Geminin cell line was generated by stable transfection using the Lipofectamine® 3000 Transfection Reagent protocol (Thermo Fisher Scientific, L3000-008). MCF-7 + p53-Venus cells were transfected with 5 μg of the pLV-EF-mCherry-Geminin IRES-Puro plasmid. After 2 days of transfection in non-selective media, cells were switched to selective media with 400 μg/ml Neomycin and 0.5 μg/ml of Puromycin for about 2 weeks to isolate the stably transfected cells.

### Determination of mCherry reporter copy number

The mCherry reporter construct copy number was determined by qPCR using the TaqMan™ Copy Number Assay protocol. A Custom TaqMan™ Copy Number Assay (Applied Biosystems 4400294) was designed to amplify the mCherry fluorescent reporter using the following primers and probes: 5′-GACCACCTACAAGGCCAAG AAG-3′ (forward primer), 5′-AGGTGATGTCCAACTTGATGTTGA-3′ (reverse primer), and 5′-6-carboxy-fluorescein(FAM)-CAGCTGCCCG GCGCCTACA-nonfluorescent quencher (NFQ)-3′ (TaqMan probe sequence). As a reference, the TaqMan® Copy Number Reference Assay RNaseP (Applied Biosystems 4403326) was run simultaneously with the custom mCherry TaqMan copy number assay. The mCherry reporter construct copy number for the clonal cell lines was calculated using the CopyCaller® Software v2.0, which confirmed that the cell lines contained one copy of the mCherry reporter construct.

### Image analysis

Customized MATLAB (2019Rb, Mathworks) code was used to process and analyze fluorescence images exported from the NIS-Elements software (Nikon; Computer Code EV1). First, image stacks were created from the fluorescence images acquired for each time-point and position upon exposure to the brightfield, mCherry, or Venus/YFP channels. Cell segmentation was manually performed for each image stack using ImageJ (NIH). The mean nuclear fluorescence intensity was measured for each cell over time by automated analysis of the image stacks at the positions determined in ImageJ (Computer Code EV1). An estimated background intensity level (i.e., the highest peak from a histogram of all pixels) was subtracted from each measurement (Computer Code EV1). Linear interpolation was performed for any frames without a measurement using measurements from surrounding frames (Computer Code EV1). For dividing cells, only one sister cell was followed to movie completion and

frames that included fluorescence spikes during division were omitted (Computer Code EV1). All single-cell traces were smoothed using the 1D Blaise filtering function within MATLAB (Computer Code EV1). To correct for changes in lamp conditions, all cell traces for each promoter reporter cell line and Nutlin-3 regimen were normalized to the average fluorescence level at the first timepoint ($t = 0$).

### Selection of cells for further characterization based on p53-Venus traces

Cells were first selected based on whether their p53-Venus trace produced the expected pulse number from the given Nutlin-3 dosing regimen. A customized MATLAB code (Computer Code EV1) was used to individually examine the p53-Venus traces for > 65 cells from each promoter reporter cell line and Nutlin-3 dosing regimen combination. Cells with p53-Venus traces that were non-responsive, contained fluorescent spikes, or had non-pulsatile responses were omitted. At least 45 cells from each promoter reporter cell line for each Nutlin-3 dosing regimen were selected for further characterization.

### Identification of responding cells based on mCherry traces

The previously selected cells (i.e., those with p53-Venus expression dynamics corresponding to the Nutlin-3 dosing regimen) were categorized based on whether their mCherry traces were responding or non-responding as follows. Customized MATLAB code (Computer Code EV1) was used to individually examine the mCherry traces for each cell from each promoter and Nutlin-3 treatment combination. Cells were categorized as non-responding if their final activation level was less than two times their minimum activation level.

### K-means clustering of promoter-mCherry reporter traces

Promoter-mCherry reporter traces for each promoter and condition were partitioned into four clusters using the *k*-means Clustering algorithm within the MATLAB Statistics and Machine Learning Toolbox (MATLAB2019a, Mathworks). *k*-Means Clustering was performed with the *correlation* distance metric and repeated five times using new initial cluster centroid positions to obtain final clusters.

### p53 amplitude modulation analysis

p53 target promoter cell lines were exposed to 5, 10, or 15 μM Nutlin-3 to modulate the p53 pulse amplitude. Changes to p53 pulse amplitude were assessed by calculating the cumulative p53 level over the 24-h movie using customized MATLAB code. Cells exposed to the Nutlin-3 conditions that produced the lowest and highest p53 levels for each promoter reporter cell line were selected for further analysis.

### Analysis of geminin-mCherry reporter traces

Cells were first selected based on whether their p53-Venus trace produced the expected pulse number from the given Nutlin-3 dosing regimen over the 40-h experiment using the previously described protocol. Selected cells were then categorized as undergoing cell cycle arrest or cell cycle progression based on the Geminin-mCherry reporter expression pattern. For cells undergoing cell cycle arrest, basal levels of Geminin-mCherry expression were observed for > 25 h of the 40-h experiment. For cells undergoing cell cycle progression, elevated levels of Geminin-mCherry expression were observed with either a pulsatile or sustained temporal pattern.

### Determination of mCherry half-lives by cycloheximide assay

MCF-7 + p53-Venus + *MDM2*p-mCherry-NLS-PEST cells were plated into a 6-well MatTek glass bottom dish (MatTek corporation) and treated with 10 μM Nutlin-3 overnight to induce mCherry expression. Following the overnight induction, Nutlin-3 was removed by washing cells three times with Transparent RPMI media containing 5% FBS and 1% antibiotic–antimycotic solution. Cells were then treated with cycloheximide (100 μg/ml) in Transparent RPMI media and immediately imaged for 24 h by time-lapse fluorescence microscopy. Images were analyzed using ImageJ, and customized MATLAB code was used to determine the half-life of the mCherry reporter protein.

### Statistical analysis

A two-sample $t$-test function (MATLAB 2015b, Mathworks) was used to determine the statistically significant differences in target promoter activation parameters (i.e., timing, magnitude, and rate) for each Nutlin-3 treatment. Differences producing a $P$-value $< 0.05$ were considered statistically significant.

## Data availability

The single-cell imaging fluorescence data and computer code for image analysis produced in this study are available in the following:

(i) Imaging data used to generate Figs 1–5 and EV1–EV5 are provided as Dataset EV1.
(ii) Imaging data used to generate Fig 6 are provided as Dataset EV2.
(iii) Computer code: Computer Code EV1.

**Expanded View** for this article is available online.

## Acknowledgements

We thank all members of the Batchelor laboratory and the Levens laboratory for thoughtful discussions. The pLV-EF-mCherry-Geminin IRES-Puro plasmid was kindly provided by S Cappell. We thank J Porter for assistance with data analysis. We thank M Aldred, J Wu, and C Corcoran for assistance with image analysis and plasmids. This work was supported by NIH grants 5R01GM124446 (to AS), 5R01GM126557 (to AS), and the Intramural Research Program of the Center for Cancer Research, National Cancer Institute, NIH project ZIA BC011382 (to EB).

## Author contributions

MDH and EB designed the project. ADB constructed plasmids and cell lines and performed preliminary experiments. MDH conducted the experiments. MDH, EB, and AS analyzed the data. WSK contributed to the image analysis. MDH and EB wrote the article.

## Conflict of interest

The authors declare that they have no conflict of interest.

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
