## [Review Process File · Molecular Systems Biology]

p53 Pulse Modulation Differentially Regulates Target Gene Promoters to Regulate Cell Fate Decisions

Marie D Harton, Woosuk Koh, Amie D Bunker, Abhyudai Singh and Eric Batchelor.

Review timeline:

Submission date:	11 th October 2018
Editorial Decision:	15 th November 2018
Revision received:	18 th July 2019
Editorial Decision:	26 th August 2019
Revision received:	29 th August 2019
Accepted:	2 nd September 2019

Editor: Maria Polychronidou

Transaction Report:

1st Editorial Decision

15th November 2018

Thank you again for submitting your work to Molecular Systems Biology. We have now heard back from the three referees who agreed to evaluate your manuscript. As you will see below, the reviewers raise substantial concerns on your work, which, unfortunately preclude its publication in Molecular Systems Biology.

The reviewers acknowledge that the addressed topic is interesting. However, they point out that the work remains too preliminary at this point and the main conclusions are not convincingly supported by the performed experimental and computational analyses. As such, they indicated that they do not support publication in Molecular Systems Biology.

Considering the substantial points raised and the overall rather low level of support provided by the reviewers, we have no choice but to return the manuscript with the message that we cannot offer to publish it.

Nevertheless, since the reviewers expressed interest in the topic of the study, we would not be opposed to considering a substantially extended manuscript based on this work, provided that the most significant points raised by the reviewers can be convincingly addressed. Some of the points that are particularly important are the need to include analyses of the role of the p53-MDM2 feedback loop, analyses of the data at the single-cell level (instead of averaged data) and further analyses using the mathematical model in order to extend the level of mechanistic and quantitative insight. The reviewers provide constructive suggestions related to these points. Importantly, a direct demonstration that the conclusions of the study extend our understanding beyond the conclusions of the studies by Porter et al, 2016 and Hafner et al, 2017 needs to be included. All other issues raised by the referees would also need to be addressed. We recognize that thoroughly addressing the referees' concerns would involve substantial further experimentation and analyses with unclear outcome, which clearly go beyond the scope of a major revision.

 REFEREE REPORTS

Reviewer #1:

Harton and colleagues investigate p53 dynamics in response to several pulsatile stimulation patterns and the effect of p53 dynamics on two canonical p53 targets, MDM2 and CDKN1A. While the premise of this work is interesting, my major concern is that the paper gives the impression of being thin and unconvincing. The paper is lacking supplementary information with data, sample movies of discussed phenotypes and explanation of the analysis. In its current form I cannot recommend this paper for publication.

Specifically:

1. The well-known p53-mdm2 feedback loop makes the results of this manuscript difficult to interpret, in particular when the authors compare the temporal dynamics of MDM2 and CDKN1A promoters.

- a) The implications of this feedback on p53 downstream promoters should be better discussed.
- b) Exogenously introduced MDM2 promoter could compete in p53 binding with the endogenous one and therefore, through the feedback, could alter p53 expression dynamics. Did the author compare p53 expression dynamics in the presence or absence of the artificial MDM2 promoter?
- c) Could the authors comment on possible differences between MDM2 promoter-mCherry and CDKN1A promoter-mCherry with respect to the competition between endogenous and artificial MDM2 promoters?

2. Data analysis and presentation

- a) Data from single-cell time-lapse experiments (Figs. 1D and 2B) are difficult to read as they are quite noisy and heterogeneous. A heatmap, organized/sorted with some clustering method would ease the interpretation
- b) In Fig.2B-C the authors identify two subpopulations. How did they do it? Manually, clustering?
- c) Instead of repeating the cartoon with timing, magnitude, and rate 4 times throughout the paper, please show actual time series with the example quantification of the 3 features. This would give an idea of the validity of the feature extraction methodology.
- d) Why do authors show single-cell data as population-averaged means? Why not to display them as boxplots + dotplots to give an idea about the variability?
- e) Experimental results in Fig.7B and C (+/- Trichostatin) look virtually the same to me, bar 4 top curves in the treated condition (panel C). Maybe a boxplot at several chosen time points would be more convincing?

3. The Geminin experiment in figure 6 is incomplete:

- a) In addition to frequency modulation, the authors should also evaluate division with respect to the modulation of the amplitude and duration
- b) Showing cell division episodes in phase contrast as in 6A is uninformative. The authors should show geminin expression with fluorescence images, in addition to phase contrast, and with a larger field of view.
- c) Images in 6A should show the different tested conditions, for comparison.
- d) What is the purpose of showing three curves in 6B? In my opinion Fig.6B and S4 are useless since cell division events are not synchronized. We suggest to show more trajectories (e.g. 50 per condition) and visualize them as heatmaps.

3. Modelling

The table in Fig.S5 is a nice summary, however, I am missing the actual results from the simulations. It would be helpful to see some of the time series predicted by the model except the two shown in the main manuscript.

4. The manuscript is redundant in several parts and same concepts are repeated many times. The presentation is relying too much on conceptual cartoons instead of including the actual data used

for quantification. Moreover, similar cartoons are repeated many times in different figures without the actual need. The authors should put more effort to make manuscript and figures more concise to improve readability.

5. No actual movies with sample phenotypes are included.

Minor concerns:

1) In S2, frequency, duration and amplitude modulations should be shown in the same order in both A-B-C and D-E-F.

Reviewer #2:

The manuscript of Harton et al. describes how varying the pulsatile dynamics (amplitude, duration and frequency) of P53 affects the activation of canonical target gene promoters (MDM2 and CDKN1A). For this the authors used stable MCF7 cell lines that express p53-Venus in which they engineered a p53 target promoter controlling expression of mCherry. P53 pulsatile dynamics was controlled by the small molecule Nutlin-3 that inhibits MDM2.

The major conclusions are that MDM2 promoter acts like a band pass filter selecting the natural P53 oscillation frequency and the CDKN1A promoter as a low pass filter. The authors show that high frequency P53 oscillations release cells from the P53 activated CDKN1A mediated p21 cell cycle arrest and relate this to the low pass filter feature of CDKN1A.

The study tries to recapitulate the promoter responses by an oversimplified model that only includes reversible P53 binding and activation of promoters.

In general, the study lacks thorough analysis of cell responses to the pulsatile P53 inputs and no clustering analysis is applied to the highly heterogeneous responses. Only averages are considered. In some cases replicates are lacking. The BP and LP features of the promoters are relatively weak and not completely consistent with the alleviation of cell cycle arrest. The mathematical modelling is oversimplified (it for example lacks the autocatalytic amplification of the MDM2 induced P53 expression). There is not much of a mechanism that the authors describe how the system acquires its frequency filtering abilities.

Reviewer #3:

The paper by Harton et al addresses a timely and important question of the decoding of signaling dynamics in mammalian cells. The encoder/decoder paradigm for signal transduction is gaining traction in the field, but good quantitative understanding of signal decoding is still missing. The paper uses a smart experimental system using microfluidics to control concentration of MDM2 inhibitor to manipulate p53 levels and analyzes how two promoters reporter genes change in response to these dynamic inputs. I found the work important and interesting and I think it should be published in MSB after some more work on presentation and analysis as detailed below.

The current presentation and the analysis / modeling presented does not fully capture the potential that this paper has and I encourage the authors to revise their work. Below are a few constructive suggestions and I hope the authors will seek more input related to these issues. I want to emphasize that I do not think that any more experimental work needs to be done and that I believe that most of the suggestions below could be looked into within less than two months.

Context and proper comparison to previous findings.

The experimental design and the use of promoter reporters is unique and allows the authors to specifically identify promoter decoding separately from mRNA half life. Two recent papers by Hafner et al and Porter et al studied the question of decoding by measuring mRNA response. The conclusion of these papers was that to a large degree everything can be explained by mRNA half-life arguing that there is no need to evoke additional regulation, in other words, the current understanding of promoter decoding of p53 dynamics is that it doesn't happen. When the authors write that: "Although recent single cell studies of p53 have demonstrated that cells leverage

differences in mRNA half-lives to induce diverse target gene responses (Hafner et al., 2017; Porter et al., 2016), the role of target promoter activation in the cellular decoding of p53 dynamics is not understood" it doesn't convey the right tone to understand how meaningful their results are. I recommend increasing the contrast in presentation and also performing additional analysis to strengthen this point. If they take some of the mRNA data from either Hafner or Porter for CDKN1A and MDM2, or even better, the MDM-YFP data they collected in this work, how much better does a complex model that include promoter decoding fit the data compared to the simpler models that don't include this. Even if the conclusion is that the effect size is small, this data is important to show.

Single cell analysis and estimation of response variability:

The authors don't utilize enough the fact that they measure everything in single cell. The data the paper uses is all single cell dynamic measurement, but none of the analysis is done by comparing single cell p53 to single cell response. Everything is done via averaging of features at classes of inputs. Given the high degree of variance in p53 dynamics and response, it is possible that the authors are losing a lot of statistical power and under report the magnitude of the differences between the two promoters. Furthermore, they lose the ability to estimate cellular variability in response and see how different are cells from each other. This can be done by estimating the "residual" after correlating p53 dynamics and promoter response or by using a modeling approach that fits the single cell data and look at distribution of kinetic parameters (see Yao et al 2016 MSB).

Math modeling:

The authors use a three state model of promoter state to interpret their experimental findings. A model like this is important as it provides a way to think mechanistically on the input/output feature usage. However, the use of the model is very limited. It is mostly shown to predict response to TSA treatment. The model should play a much more prominent role and really help the reader understand what degree of parameter changes are needed to get the differences between MDM2 and CDKN1A. Furthermore, the model can be used to answer a more general question, what is the limit of promoter decoding differences? Are the differences between the pair CDKN1A and MDM2 are the biggest one should expect? Also the model should be used to compare the effect size of the promoter decoding parameters to mRNA half-life parameters. Maybe figure S5 is doing some of that, but 1) not enough as there is no comparison to mRNA lifetime, 2) it is pretty well hidden, 3) the way that the data is presented in that figure is not ideal.

Confusion related to timescales:

The half life of the reporter plays a key role in interpreting their data, yet the information on that was somewhat hidden in the model fitting part. Ideally, the authors should have used a reporter with a lifetime that is shorter than the pulse period. This type of reporter would not have integrated information about pulses. Unfortunately, the mCherry-PEST that was used had a measured half-life of 7.5 hours, so it is capable of integrating information between peaks. I am not proposing that the authors redo all the experiment in this paper, but to explain visually what is the expected promoter reporter from different dynamic input given this lifetime. Also, the mRNA lifetime used by the authors seems very short (4 minutes!) either this is a typo or a value that is simply incorrect. As the value matters for their model, presentation and the comparison with models that are based on mRNA decay, I strongly suggest that they look more into it.

Presentation:

At its current form the paper reads a bit like a laundry list of input dynamics features that change / don't change another list of output features in response of the two chosen target promoters. Figure 1 and 2 set things up and figures 2-5 are going over each of these input features. Figure 6 is dedicated to show some broader relevance by showing how these feature impact cell cycle. Figure 7 proposes a model with some minimal prediction and validation whereas figure 8 again summarizes the whole thing. I would recommend to shrink the length dedicated to cataloging relationship between input/output features and increase the analysis and discussion of how it actually works. Figures 1+2 could easily be merged, so do figures 3-5. Figure 8 could potentially be removed.

Small typo: There is a typo in the equation about p53 dynamics as a sum of gaussians and the denominator should be $2s^2$ and not $2t^2$.

Thank you for considering the revised version of our manuscript entitled “p53 Pulse Modulation Differentially Regulates Target Gene Promoters to Regulate Cell Fate Decisions” (manuscript reference # MSB-18-8685). We appreciate the careful reading of our manuscript by the editors and the reviewers, and we are grateful for their feedback. The reviewers raised several thoughtful concerns about the results we presented in our original manuscript. We have worked hard to address all of the concerns raised by the reviewers through additional experiments, further computational analysis of our original data, and more detailed discussion of our findings. As a result, in this version of the manuscript we have updated all main figures, and we have included several new supplementary figures and movies. We have also revised the text to describe our new experimental and computational results, to clarify several points that the reviewers found unclear, and to elaborate on the scope and implications of our findings. We believe that addressing the concerns of the reviewers has helped us to significantly improve the manuscript.

Below, please find a point-by-point response to the reviewers' comments describing how we have addressed their concerns.

We once again thank you and the reviewers for your comments. We hope that you will find the revised manuscript suitable for publication in *Molecular Systems Biology*.

Please do not hesitate to contact me if you have any comments or concerns. We look forward to hearing from you.

Sincerely,
Eric Batchelor, PhD
Investigator, Laboratory of Cell Biology, NCI

Reviewer #1:

1. The well-known p53-mdm2 feedback loop makes the results of this manuscript difficult to interpret, in particular when the authors compare the temporal dynamics of MDM2 and CDKN1A promoters.

a) The implications of this feedback on p53 downstream promoters should be better discussed.

The reviewer is correct that p53 and MDM2 form a well-known and well-characterized negative feedback loop, as p53 upregulates *MDM2* expression and MDM2 tags p53 for degradation. Our strategy of using Nutlin-3 actually relies on this negative feedback loop: Nutlin-3 binds to MDM2 to disrupt the interaction of MDM2 with p53, enabling p53 to accumulate. It is precisely because of the negative feedback between these two proteins and Nutlin-3's ability to disrupt it that we are able to manipulate the dynamics of p53 expression. For our analysis, we are focusing on p53 expression as the “input” of the genetic circuit, and by using Nutlin-3 we can precisely define the input. We have explained this strategy, including the role of the negative feedback loop in enabling this strategy, in more detail in the current manuscript.

b) Exogenously introduced MDM2 promoter could compete in p53 binding with the endogenous one and therefore, through the feedback, could alter p53 expression dynamics. Did the author compare p53 expression dynamics in the presence or absence of the artificial MDM2 promoter?

Several CHIP-seq studies have identified greater than 4,000 p53 binding sites in the genome (Bao et al., 2017; Hafner et al., 2017; Verfaillie et al., 2016). Our transcriptional reporters of MDM2 and CDKN1A activation add a single additional p53 binding site to the genome; thus, we expected this additional site would provide minimal disruption to endogenous p53 expression dynamics. To verify this expectation, as suggested by the reviewer, we compared p53-Venus levels between MCF7 cells expressing either p53-Venus alone or p53-Venus and the *MDM2* promoter-mCherry construct. There was no significant change in p53 levels when the *MDM2* promoter construct was present. We have added a supplemental figure showing these results.

c) Could the authors comment on possible differences between MDM2 promoter-mCherry and

CDKN1A promoter-mCherry with respect to the competition between endogenous and artificial MDM2 promoters?

This concern appears to be related to a combination of the previous two comments. The single exogenous *MDM2*-promoter we added to our reporter cell line (containing >4000 endogenous p53 binding sites) showed no significant alteration to p53-Venus expression. Thus, we do not expect the differences in *MDM2* promoter-mCherry and *CDKN1A* promoter-mCherry responses that we identified are due to competition effects between endogenous and exogenous *MDM2*-promoter p53 binding.

2. Data analysis and presentation

a) Data from single-cell time-lapse experiments (Figs. 1D and 2B) are difficult to read as they are quite noisy and heterogeneous. A heatmap, organized/sorted with some clustering method would ease the interpretation.

We compared data presentation formats (i.e., individual traces or heatmaps) to best observe the heterogeneity in p53-Venus and mCherry expression levels. For p53-Venus pulsatile inputs, we found the individual traces to better display the changes in pulse behaviors when compared to the heatmap format. However, as suggested by the reviewer, we now also include heat maps for p53-Venus and mCherry expression for all conditions, as different readers may prefer different formats for data visualization. In addition, as suggested by the reviewer, we performed k-means clustering of mCherry single cell traces to better characterize the heterogeneous response profiles for each promoter and p53 pulse modulated input. We included a description of the clustering in the main text and visualization of the clusters as a supplemental figure.

b) In Fig.2B-C the authors identify two subpopulations. How did they do it? Manually, clustering?

As described above, we now present all traces in multiple formats, including clustering, and we describe in greater detail how the single cell data are represented in the revised manuscript.

c) Instead of repeating the cartoon with timing, magnitude, and rate 4 times throughout the paper, please show actual time series with the example quantification of the 3 features. This would give an idea of the validity of the feature extraction methodology.

We thank the reviewer for their suggestions to help improve our data presentation. We have now extensively revised how we present our results in each of the figures to reduce repetition and improve clarity.

d) Why do authors show single-cell data as population-averaged means? Why not to display them as boxplots + dotplots to give an idea about the variability?

We again thank the reviewer for their suggestion regarding data presentation. To better highlight the single cell nature of our data, we now present results as dotplots with boxplots, which enables better visualization of the inherent variability in all responses.

e) Experimental results in Fig.7B and C (+/- Trichostatin) look virtually the same to me, bar 4 top curves in the treated condition (panel C). Maybe a boxplot at several chosen time points would be more convincing?

We agree that the results of the trichostatin A experiments were potentially difficult to interpret. We have now removed these experiments from the manuscript, as we do not believe they added a significant contribution to our study.

3. The Geminin experiment in figure 6 is incomplete:

a) In addition to frequency modulation, the authors should also evaluate division with respect to the modulation of the amplitude and duration.

We thank the reviewer for this helpful suggestion. We initially focused on the effects of frequency modulation on the cell cycle due to the identification of the *CDKN1A* promoter as a low-pass filter. We showed that the filtering occurred not only at the level of promoter activation but was also

maintained in the actual cell cycle arrest phenotype. At the reviewer's suggestion, we also analyzed how p53 pulse amplitude and pulse duration affect cell cycle arrest. Consistent with our results showing that the *CDKN1A* promoter is more sensitive to p53 pulse amplitude modulation than duration modulation, we found that cell cycle arrest was also more dependent on p53 pulse amplitude than duration. We have included these new results in Fig. S6B and describe them in the text.

b) Showing cell division episodes in phase contrast as in 6A is uninformative. The authors should show geminin expression with fluorescence images, in addition to phase contrast, and with a larger field of view.

and

c) Images in 6A should show the different tested conditions, for comparison.

We now include bright field, p53-Venus fluorescence, and geminin-mCherry fluorescence images of a larger field of cells in Fig. 6A, Fig. 6B, and Fig S6A for the low, high, and natural frequency p53 pulse experiments, respectively.

d) What is the purpose of showing three curves in 6B? In my opinion Fig.6B and S4 are useless since cell division events are not synchronized. We suggest to show more trajectories (e.g. 50 per condition) and visualize them as heatmaps.

As suggested, in Fig 6B we now present the traces of geminin-mCherry expression for all tracked cells for low, natural, and high frequency conditions as heat maps.

3. Modelling

The table in Fig.S5 is a nice summary, however, I am missing the actual results from the simulations. It would be helpful to see some of the time series predicted by the model except the two shown in the main manuscript.

We have made significant changes to our modeling in the revised version of this manuscript. We developed a simple model of promoter activation which we fit using biochemical parameters we measured explicitly, as well as previously quantified parameters from the literature. We also fit our single cell data to our model, as presented in Fig. S4 for representative single cells with the MDM2 promoter reporter and in Fig. 3 as a summary across single cell traces for both promoters. We present a detailed description of the modeling in the main text, including the empirically fitted parameters for each promoter, with additional details in the revised Methods section of the manuscript.

4. The manuscript is redundant in several parts and same concepts are repeated many times. The presentation is relying too much on conceptual cartoons instead of including the actual data used for quantification. Moreover, similar cartoons are repeated many times in different figures without the actual need. The authors should put more effort to make manuscript and figures more concise to improve readability.

As suggested, we have removed the repeated cartoons and included single cell data. We have also revised the text of the manuscript to be less redundant when discussing characterization of the promoter-mCherry reporters to better highlight the biological insight gained from our study.

5. No actual movies with sample phenotypes are included.

As suggested, we have now included several movies in the supplemental information.

Minor concerns:

1) In S2, frequency, duration and amplitude modulations should be shown in the same order in both A-B-C and D-E-F.

In this revised manuscript, we have maintained consistency in the presentation of results across figures.

Reviewer #2:

1. In general, the study lacks thorough analysis of cell responses to the pulsatile P53 inputs and no clustering analysis is applied to the highly heterogeneous responses. Only averages are considered. In some cases replicates are lacking.

In our revised manuscript, we have now applied k-means clustering to better characterize the heterogeneous *MDM2* and *CDKN1A* promoter responses (new Fig. S3). We have also included heatmaps sorted based on integrated p53-Venus or mCherry fluorescence levels to improve the data presentation (Figs. 1, 2, 6, S1). As suggested by Reviewer 1, we also present jittered boxplot formats and not averages to show single cell response variability (see Figs. 4, 5, S5). All experiments have been performed with biological replicates, and to improve the number of cells analyzed we have now included analysis of promoter responses for all cells and not just the “responding” cells as was done in the original manuscript.

2. The BP and LP features of the promoters are relatively weak and not completely consistent with the alleviation of cell cycle arrest.

We have performed statistical analysis and determined that the reduction in *MDM2* promoter activation for low and high frequency are significant with respect to the natural frequency response. Similarly, the reduction in *CDKN1A* promoter activity is significantly reduced in comparison to the low and natural frequency promoter response. Analysis of cell cycle progression via single cell tracking of geminin-mCherry expression as well as cell division events was consistent with the *CDKN1A* promoter responses to frequency modulation. In this revised version of the manuscript, as suggested by Reviewer 1, we also analyzed the effects of amplitude and duration modulation on the cell cycle (Fig. S6), finding a clear correlation between *CDKN1A* promoter responses and cell cycle arrest in all modulations.

3. The mathematical modelling is oversimplified (it for example lacks the autocatalytic amplification of the MDM2 induced P53 expression).

We acknowledge that our model does not take into account all known interactions in the p53 network. Instead, the goal of our modeling was to provide a quantitative framework that could aid in the synthesis and interpretation of our experimental results. To this end, we employed a relatively simple model of promoter activation that provided several new insights into the function of p53 dynamics in promoter activation, including: quantifying the differences in maximal activation for the two promoters in individual cells; identification of a comparable, large Hill coefficient indicating a fairly sharp transition to an active state for both promoters; and quantifying the difference in the amplitude of p53 levels required to activate the two promoters. Regarding modeling the interactions between MDM and p53, as we discussed in our response to Reviewer 1, our experimental design was developed such that we could make use of Nutlin-3 dosing regimens to generate specific p53 dynamical responses through inhibition of *MDM2* activity. Through such a method, we could directly use the p53-Venus expression in individual cells as the direct “input” for our modeling, allowing us the freedom to remain fairly agnostic to how the “input” was mechanistically achieved on a cell-by-cell basis, and thus direct our studies to address how p53 dynamics directly impact promoter activation. Future studies will be able to continue to build on our current model iteratively, as necessary, to increase our understanding of the complex p53 network.

4. There is not much of a mechanism that the authors describe how the system acquires its frequency filtering abilities.

We now include a discussion of potential mechanisms generating the bandpass and lowpass filtering of p53 pulse frequency in the Discussion of our revised manuscript. The novel observation of dynamic filtering of p53 frequency that we have quantified is only one aspect of the biological impact of p53 dynamics in our study. In addition to the dynamic filtering of frequency, we show here the thresholding of promoter activation that occurs as a function of p53 amplitude, as well as a refractory period that arises with long duration p53 pulses as a function of *MDM2* levels. Although we speculate about potential mechanisms by which the dynamic filtering can occur, we feel that work detailing the molecular mechanisms by which the filters arise for each promoter is beyond the scope of this manuscript.

Reviewer #3:

The paper by Harton et al addresses a timely and important question of the decoding of signaling dynamics in mammalian cells. The encoder/decoder paradigm for signal transduction is gaining traction in the field, but good quantitative understanding of signal decoding is still missing. The paper uses a smart experimental system using microfluidics to control concentration of MDM2 inhibitor to manipulate p53 levels and analyzes how two promoters reporter genes change in response to these dynamic inputs. I found the work important and interesting and I think it should be published in MSB after some more work on presentation and analysis as detailed below.

The current presentation and the analysis / modeling presented does not fully capture the potential that this paper has and I encourage the authors to revise their work. Below are a few constructive suggestions and I hope the authors will seek more input related to these issues. I want to emphasize that I do not think that any more experimental work needs to be done and that I believe that most of the suggestions below could be looked into within less than two months.

We thank the reviewer for their support of our manuscript.

1. Context and proper comparison to previous findings.

The experimental design and the use of promoter reporters is unique and allows the authors to specifically identify promoter decoding separately from mRNA half life. Two recent papers by Hafner et al and Porter et al studied the question of decoding by measuring mRNA response. The conclusion of these papers was that to a large degree everything can be explained by mRNA half-life arguing that there is no need to evoke additional regulation, in other words, the current understanding of promoter decoding of p53 dynamics is that it doesn't happen. When the authors write that: "Although recent single cell studies of p53 have demonstrated that cells leverage differences in mRNA half-lives to induce diverse target responses (Hafner et al., 2017; Porter et al., 2016), the role of target promoter activation in the cellular decoding of p53 dynamics is not understood" it doesn't convey the right tone to understand how meaningful their results are. I recommend increasing the contrast in presentation and also performing additional analysis to strengthen this point. If they take some of the mRNA data from either Hafner or Porter for CDKN1A and MDM2, or even better, the MDM-YFP data they collected in this work, how much better does a complex model that include promoter decoding fit the data compared to the simpler models that don't include this. Even if the conclusion is that the effect size is small, this data is important to show.

We thank the reviewer for their thoughts on how our work described in this manuscript relates to the larger literature, as well as their appreciation of the novel results we describe. At the reviewer's suggestion, we have rewritten the text to better highlight the contrast between the previous work from our lab and from Hafner et al. focusing on target mRNA decay rates as a discriminator of dynamical responses in the p53 network, and our new results here showing that promoter responses to p53 dynamics also give rise to additional emergent properties. Of particular note, the two target genes on which we focus here, *MDM2* and *CDKN1A*, have very comparable mRNA half-lives of 2.66 h and 2.79 h, respectively. We now highlight this fact in Fig. 1A of the revised manuscript. However, as we find in our results in this manuscript, even though the mRNA decay rates are comparable, the responses at the level of promoter activation differ greatly, and these differences result in phenotypic changes in the cellular responses regulated by p53 dynamics.

With regard to further analysis to highlight the contrast in promoter activation versus mRNA decay rates and their roles in MDM2 and p21 regulation, in our revised manuscript we have used our single cell data to directly construct the "dose-response curves" for promoter activation as a function of p53 levels (Figure 3). With our fitting to even this relatively simple promoter model, we found distinct thresholds and maximal levels of activation of the two promoters, empirically identifying how differential decoding by the promoters can occur independent of mRNA decay rates. We have included a discussion of these results and also place our findings into the context of new results on variability of cell cycle regulation as a function of p53 amplitude (Reyes, et al. *Mol Cell* 2018).

2. Single cell analysis and estimation of response variability:

The authors don't utilize enough the fact that they measure everything in single cell. The data the paper uses is all single cell dynamic measurement, but none of the analysis is done by comparing

single cell p53 to single cell response. Everything is done via averaging of features at classes of inputs. Given the high degree of variance in p53 dynamics and response, it is possible that the authors are losing a lot of statistical power and under report the magnitude of the differences between the two promoters. Furthermore, they lose the ability to estimate cellular variability in response and see how different are cells from each other. This can be done by estimating the "residual" after correlating p53 dynamics and promoter response or by using a modeling approach that fits the single cell data and look at distribution of kinetic parameters (see Yao et al 2016 MSB).

We thank the reviewer for this suggestion, which was similar to a concern raised by Reviewer 1. In our initial manuscript, we wanted to focus on the novel insights into p53 regulation that we identified, including amplitude-dependent threshold, a duration-dependent refractory period, and frequency-dependent dynamic filtering. We perhaps erred on the side of over-simplification of data presentation in our initial manuscript. We have extensively revised our manuscript to better highlight the wealth of data we have quantified from single cells. For example, we now include calculation of single cell p53-promoter activation “dose response curves” (Fig. S4), k-means clustering analysis of individual cellular responses (Fig. S3), presentation of all single cell p53 traces, promoter activation traces, and geminin traces throughout the manuscript, as well as explicit measurements of variability in multiple promoter activation characteristics (Figs. 4-5). We have performed appropriate statistical analysis of these data to ensure that we are focused on significant differences.

The studies presented in even the original version of the manuscript represent a large amount of data, as Reviewer 3 has implied when stating, “I do not think that any more experimental work needs to be done.” In this revised manuscript, we have actually included even more experimental data to support our findings. We have found that these data sets have provided a wealth of new information regarding promoter decoding of p53 dynamics, which is helping to guide some of our future studies in addition to the conclusions presented here. It is our hope that our single cell data will prove a valuable resource for not only our own work, but also the future work of the greater scientific community.

3. Math modeling:

The authors use a three state model of promoter state to interpret their experimental findings. A model like this is important as it provides a way to think mechanistically on the input/output feature usage. However, the use of the model is very limited. It is mostly shown to predict response to TSA treatment. The model should play a much more prominent role and really help the reader understand what degree of parameter changes are needed to get the differences between MDM2 and CDKN1A. Furthermore, the model can be used to answer a more general question, what is the limit of promoter decoding differences? Are the differences between the pair CDKN1A and MDM2 are the biggest one should expect? Also the model should be used to compare the effect size of the promoter decoding parameters to mRNA half-life parameters. Maybe figure S5 is doing some of that, but 1) not enough as there is no comparison to mRNA lifetime, 2) it is pretty well hidden, 3) the way that the data is presented in that figure is not ideal.

We agreed with the Reviewer in that the modeling approach we used in our original manuscript was limited, especially since it was used to focus on the effects of TSA treatment in the promoter responses. In revising the manuscript, we took a step back and decided to focus on the main findings of our study: p53 amplitude-dependent thresholding, duration-dependent refractory period, and frequency-dependent dynamic filtering. In this study, we found that a more focused modeling approach could be used to empirically characterize one of these main findings, the amplitude-dependent thresholding. We were able to compute the threshold levels of p53, as well as the cooperativity and the maximal promoter activity, for the two different promoters. In this way, we were able to address the Reviewer’s concern of identifying which parameter changes were different between the two promoters – the maximal promoter activity and the p53 threshold. We have included the new modeling results in the text and in a new Fig. 3. Regarding the Reviewer’s suggestion of whether the observed differences between the *MDM2* and *CDKN1A* promoter are the largest possible, from our revised, more-focused model it is clear that there is a broad range of promoter activation behavior that can occur as a function of p53 threshold and maximal transcriptional rate. In fact, previous studies have show that *MDM2* and *CDKN1A* are some of the most responsive p53 target genes. It is intriguing that even for these highly active targets, we have identified distinct differences in the decoding of p53 dynamics at the promoter level.

4. Confusion related to timescales:

The half life of the reporter plays a key role in interpreting their data, yet the information on that was somewhat hidden in the model fitting part. Ideally, the authors should have used a reporter with a lifetime that is shorter than the pulse period. This type of reporter would not have integrated information about pulses. Unfortunately, the mCherry-PEST that was used had a measured half-life of 7.5 hours, so it is capable of integrating information between peaks. I am not proposing that the authors redo all the experiment in this paper, but to explain visually what is the expected promoter reporter from different dynamic input given this lifetime.

As the Reviewer suggests, we include a cartoon of expected fluorescent protein output for the synthetic transcriptional reporters in Fig. 1C. Although the mCherry-PEST protein has a longer half-life than, for example, MDM2, we experimentally quantified the mCherry-PEST half-life in our experimental conditions and used the parameter explicitly in our revised modeling presented in Fig. 3. Our model fitting focused on the first few peaks of p53 oscillations, which enabled us to circumvent much of the potential concern of using a fluorescent read-out with a longer half-life.

5. Also, the mRNA lifetime used by the authors seems very short (4 minutes!) either this is a typo or a value that is simply incorrect. As the value matters for their model, presentation and the comparison with models that are based on mRNA decay, I strongly suggest that they look more into it.

We thank the reviewer for the careful reading of our manuscript. The mRNA lifetime originally reported was a typo, and it has been corrected in the revised manuscript.

6. Presentation:

At its current form the paper reads a bit like a laundry list of input dynamics features that change / don't change another list of output features in response of the two chosen target promoters. Figure 1 and 2 set things up and figures 2-5 are going over each of these input features. Figure 6 is dedicated to show some broader relevance by showing how these feature impact cell cycle. Figure 7 proposes a model with some minimal prediction and validation whereas figure 8 again summarizes the whole thing. I would recommend to shrink the length dedicated to cataloging relationship between input/output features and increase the analysis and discussion of how it actually works. Figures 1+2 could easily be merged, so do figures 3-5. Figure 8 could potentially be removed.

This concern was similar to one raised by Reviewer 1. In the revised manuscript, we have made great efforts to improve the clarity of the data presentation, and we have removed many of the redundant cartoons. As suggested by the reviewer, we have also removed Figure 8 as being unnecessary. We thank the reviewer for all of their helpful suggestions, and we believe the manuscript now presents our findings in a much easier format that allows for a greater focus on biological insights gained from our study.

7. Small typo: There is a typo in the equation about p53 dynamics as a sum of gaussians and the denominator should be $2s^2$ and not $2t^2$.

Although we again thank the reviewer for the careful reading of the manuscript, this computational analysis has been revised in the current manuscript and the equation is no longer present.

References in Response to Reviewers' Comments

Bao, F., LoVerso, P.R., Fisk, J.N., Zhurkin, V.B., and Cui, F. (2017). p53 binding sites in normal and cancer cells are characterized by distinct chromatin context. *Cell Cycle* 16, 2073-2085.

Hafner, A., Lahav, G., and Stewart-Ornstein, J. (2017). Stereotyped p53 binding tuned by chromatin accessibility. *bioRxiv*, 177667.

Reyes, J., Chun, J.-Y., Stewart-Ornstein, J., Karhohs, K.W., Mock, C.S., and Lahav, G. (2018). Fluctuations in p53 signaling allow escape from cell-cycle arrest. *Mol Cell* 71, 581-591.

Verfaillie, A., Svetlichnyy, D., Imrichova, H., Davie, K., Fiers, M., Kalender Atak, Z., Hulselmans, G., Christiaens, V., and Aerts, S. (2016). Multiplex enhancer-reporter assays uncover unsophisticated TP53 enhancer logic. *Genome Research* 26, 882-895.

2nd Editorial Decision

26th August 2019

Thank you again for sending us your revised study. We have now heard back from the two referees who were asked to evaluate your study. The same reviewers (#1 and #3) had evaluated the previous version of your work. As you will see below, the reviewers think that the study has improved as a result of the revisions and additional analyses performed, and they are now supportive of publication in *Molecular Systems Biology*.

REFeree REPORTS

Reviewer #1:

The manuscript has improved significantly and all of my concerns have been addressed in a satisfactory manner.

Reviewer #3:

The authors sufficiently addressed all the issues raised by myself and other reviewers. I recommend that the paper is published as is without delay.

2nd Revision - authors' response

29th August 2019

Reviewer comments:

Reviewer #1:

The manuscript has improved significantly and all of my concerns have been addressed in a satisfactory manner.

Reviewer #3:

The authors sufficiently addressed all the issues raised by myself and other reviewers. I recommend that the paper is published as is without delay.

We thank both of the reviewers for their enthusiastic support of our manuscript.

Accepted

2nd September 2019

Thank you again for sending us your revised manuscript. We are now satisfied with the modifications made and I am pleased to inform you that your paper has been accepted for publication.

Corresponding Author Name: Eric Batchelor

Manuscript Number: MSB-18-8685RR